# EGG: An Expert-Guided Agent Framework for Kernel Generation

**Yaochen Han** [* 1]  **Ke Fan** [* 2]  **Hongxu Jiang** [1]  **Wanqi Xu** [1]  **Weiyu Xie** [3 4]  **Runhua Zhang** [1]  **Chenhui Zhu** [1]  **Yixiang Zhang** [1]

## Abstract

High-performance GPU kernels are critical for reducing the exponentially growing computational costs of large language models (LLMs), but their development heavily relies on manual tuning by domain experts. While recent advances in LLM-based approaches show promise for automating kernel generation, they still struggle to achieve both correctness and high performance. This limitation primarily arises from the lack of domain-specific optimization guidance, hindering effective exploration of the optimization space. We propose **EGG**, an Expert-Guided Agent Framework for Kernel Generation, which incorporates expert optimization principles to guide LLMs' decisions. Inspired by expert workflows, we decompose kernel generation into two hierarchical stages: 1) algorithmic structure design, which establishes a high-quality computational structure foundation; 2) hardware-specific tuning, which performs targeted adjustments through parallel mapping, tensor tiling, and memory optimization. This staged decomposition defines explicit optimization objectives, structuring the design space to achieve progressive refinement. To this end, a stage-aware multi-agent collaboration mechanism is designed for inter and intra-stage context management, ensuring stable optimization trajectories. Experiments on KernelBench and real-world workloads show that EGG achieves a $2.13\times$ average speedup over PyTorch, outperforming existing agent-based and RL-based approaches.

## 1. Introduction

Large language models (LLMs) drive increasing training and inference costs(Raiaan et al., 2024). High-performance GPU kernels are critical to reducing these costs by determining the throughput and efficiency of modern deep learning workloads. However, developing efficient GPU kernels still relies heavily on manual tuning by domain experts. As GPU and model architectures evolve and diversify, this process becomes increasingly expensive and time-consuming (Chen et al., 2018). These limitations motivate automated GPU kernel generation as a critical research problem.

Recent advances in LLMs have demonstrated strong capabilities in general-purpose code generation (e.g., Python)(He et al., 2025; Zhang et al., 2024), making them promising candidates for automating GPU kernel generation(Jiang et al., 2026). However, unlike general-purpose programming, GPU kernel generation is a tightly constrained optimization problem over a vast, hardware-dependent design space. Kernel code must satisfy strict syntactic and semantic correctness requirements while simultaneously achieving high performance. Additionally, kernel performance depends on intricate interactions among parallel mapping, memory hierarchy utilization, and other hardware-specific features, where even minor code changes can lead to orders-of-magnitude performance differences. Prior work shows that even a small neural network subgraph can expose an optimization space of up to $10^9$ configurations on GPUs (Zhai et al., 2024). As a result, directly applying LLMs without additional information often produces kernels that are invalid or far from optimal, motivating specialized methodology to bridge this gap.

Existing LLM-based GPU kernel generation methods can be divided into two categories: 1) using fine-tuning and reinforcement learning (RL) to adapt LLMs to the kernel generation domain (Li et al., 2025c; Woo et al., 2025); 2) building agent-based systems on general-purpose LLMs, which leverage iterative refinement to improve generated kernels without additional model training(Zhang et al., 2025b; Li et al., 2025a; Zhang et al., 2025a).

Despite their differences, both classes of approaches are fundamentally constrained by the extreme complexity of the GPU kernel optimization space. 1) High-quality kernel

---
[*]Equal contribution  [1]Beihang University, Beijing, China [2]Shanghai Jiao Tong University, Shanghai, China [3]Tsinghua University, Beijing, China [4]Approaching.AI, Beijing, China. Correspondence to: Runhua Zhang <rhzhang20@buaa.edu.cn>.

*Proceedings of the 43$^{rd}$ International Conference on Machine Learning*, Seoul, South Korea. PMLR 306, 2026. Copyright 2026 by the author(s).

datasets are scarce, and generated kernels must satisfy strict syntactic, semantic, and hardware-specific constraints. As a result, RL-generated kernels may fail to compile or deliver limited performance. For example, AutoTriton reports an average correctness rate below 50% (Li et al., 2025c). 2) Agent-based approaches, such as CudaForge (Zhang et al., 2025b), improve kernel correctness and robustness via multi-turn optimization. However, most existing agents (Wang et al., 2025) lack domain-specific optimization guidance and rely on coarse-grained feedback (e.g., execution time), resulting in trial-and-error exploration and only marginal improvement after multiple optimization rounds.

These limitations suggest that fully unlocking the potential of LLMs for GPU kernel generation requires integrating expert-level kernel optimization principles to constrain the design space and guide the exploration process. Rather than relying on trial-and-error or end-to-end learning, effective systems must impose structured optimization objectives on the design space that reflect expert reasoning and hardware-aware constraints.

To this end, we propose EGG, an expert-guided agent framework for automatic GPU kernel generation. EGG guides LLMs to make stage-wise optimization decisions by explicitly modeling expert optimization workflows, enabling structured exploration of the kernel optimization space while preserving the flexibility of general-purpose LLMs.

Specifically, EGG employs **expert-guided staged optimization**, decomposing kernel generation into two hierarchical stages: **1) algorithmic structure design** and **2) hardware-specific tuning**. The algorithmic structure design establishes a strong performance upper bound by combining multi-seed search with algorithmic refinement techniques. The hardware-specific tuning then systematically realizes this potential through three sequential sub-stages with explicit objectives: parallel mapping, tensor tiling, and memory optimization. By decomposing the complex optimization problem into constrained sub-problems aligned with expert workflows, EGG effectively guides LLM-based agents toward high-performance kernel implementations.

To support this process, we propose a **stage-aware multi-agent collaboration mechanism**, which consists of a code agent, a profile agent, and a debug agent, with context management during inter and intra-stages. For inter-stage, a selective context propagation strategy retains only finalized optimization decisions while discarding intermediate outputs. For intra-stage, these agents coordinate around clearly defined objectives via structured information exchange, including bottleneck identification and optimization proposals. This design enables stable, cumulative performance improvements throughout the optimization process.

We systematically evaluate EGG on KernelBench(Ouyang et al., 2025) and real-world workloads. Experimental results show that EGG consistently generates correct and competitive kernels even for challenging tasks. EGG achieves a $2.13\times$ average speedup over PyTorch, outperforming existing agent-based and RL-based approaches.

The key contributions of this paper are as follows:

- We propose an expert-guided agent framework for GPU kernel generation, which incorporates expert kernel optimization principles to guide LLMs' decisions.

- We decompose kernel generation into *algorithmic structure design* and *hardware-specific tuning* stages according to expert workflows, which structures the optimization space to achieve higher performance.

- We design a *stage-aware multi-agent collaboration* mechanism that enables stable and cumulative performance improvements across all optimization stages, achieving 100% correctness and a $2.13\times$ average speedup on KernelBench and real-world workloads.

## 2. Related Work

### 2.1. GPU Kernel Optimization

GPU kernel optimization relies on a set of widely applicable techniques: 1) *operator fusion* reduces kernel launch overhead and redundant memory accesses by merging multiple operations(Jia et al., 2019); 2) *parallel mapping* determines how computation is distributed across the GPU execution grid, directly affecting the utilization of streaming multiprocessors (SMs)(Osama et al., 2023); 3) *tiling* partitions computation into tiles that fit register and cache capacities to maximize data reuse(Cai et al., 2023); 4) *pipelining* overlaps computation with data movement to hide memory access latency(Cheng et al., 2025).

While these optimization principles are largely shared across GPU platforms, their concrete implementations are highly hardware-dependent: SM configuration, register file size, on-chip memory hierarchy, and memory bandwidth all critically influence optimal design choices. High-performance kernels today predominantly come from expert-optimized libraries (e.g., cuBLAS(NVIDIA, 2026a), cuDNN(NVIDIA, 2026b)) or specialized implementations (e.g., FlashAttention(Dao et al., 2022)), achieving near-peak performance but at high development cost and limited adaptability to new operator structures.

### 2.2. Compilers and Triton

Compilers and domain-specific languages (DSLs) partially alleviate these challenges by providing higher-level abstractions while delegating low-level details to the com-

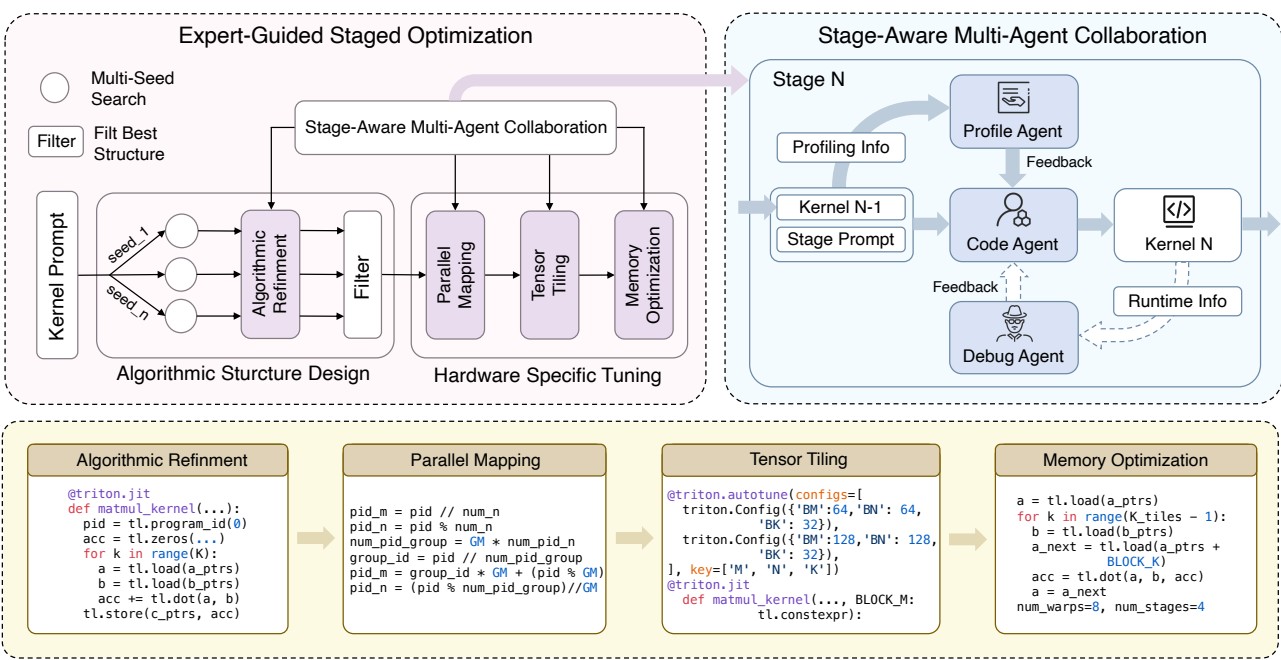

*Figure 1.* **Overview of EGG**. EGG adopts an **expert-guided staged optimization** strategy that consists of **Algorithmic Structure Design** and **Hardware-Specific Tuning** (purple region). Within each stage, a **stage-aware multi-agent collaboration** mechanism is employed to ensure stable optimization trajectories (blue region). The yellow region presents representative Triton code snippets that concretely illustrate the effect of each optimization stage. Best viewed in color.

piler(Wang et al., 2026; Spector et al., 2025). Triton (Tillet et al., 2019), a Python-embedded DSL with tile-based programming abstraction, has been widely adopted in modern LLM systems for implementing performance-critical kernels (e.g., SGLang(Zheng et al., 2024), vLLM(Kwon et al., 2023)). However, achieving high performance with Triton still requires careful manual tuning of parallelism, tiling, and pipelining decisions. Suboptimal configurations can lead to substantial performance loss. While CUDA offers finer-grained control, its larger optimization space makes automated optimization harder to control. As a result, this work focuses on Triton, as its structured abstraction provides a more tractable space for LLM-driven optimization.

### 2.3. LLM-based Kernel Generation

**Reinforcement Learning–based Methods** Pretrained LLMs struggle to generate high-performance GPU kernels due to limited understanding of hardware characteristics and optimization principles. To address this, recent works (Baronio et al., 2026; Li et al., 2025c; Woo et al., 2025; Li et al., 2026; Su et al., 2025; Fisches et al., 2025) apply task-specific fine-tuning and reinforcement learning (RL). Kevin-32B(Baronio et al., 2026) improves kernel correctness and performance through multi-round RL training. AutoTriton(Li et al., 2025c) and TritonRL(Woo et al., 2025) apply supervised fine-tuning to learn Triton syntax, then use RL to further refine performance. However, these ap-

proaches require large-scale domain-specific datasets and substantial computational resources, which may limit their practical applicability.

**Agent-based Methods** Multi-agent systems (Zhang et al., 2025b; Lange et al., 2025; Wang et al., 2025; Li et al., 2025a; Sereda et al., 2025)offer an alternative approach that avoids expensive training costs. CudaForge(Zhang et al., 2025b) employs a hardware-feedback-driven framework composed of a Coder and a Judger, using NVIDIA Nsight Compute (NCU) metrics to guide optimization. AI CUDA Engineer(Lange et al., 2025) uses stochastic mutation and iterative search to explore kernel designs. However, these methods lack a holistic understanding of kernel optimization principles and rely on coarse-grained performance feedback to guide adjustments, leading to unstable behavior and limited improvements.

## 3. Method

### 3.1. Overview

In this section, we introduce **EGG**, an expert-guided multi-agent framework for automatic generation of high-performance GPU kernels. As illustrated in Figure 1, EGG consists of two core components: 1) an *expert-guided staged optimization* strategy that decomposes the kernel optimization space into well-defined sub-problems, and 2) a *stage-aware multi-agent collaboration* mechanism that coordi-

nates specialized agents within each stage while propagating refined decisions across stages. We first describe the expert-guided staged optimization strategy, followed by the design of the stage-aware multi-agent collaboration mechanism for each optimization stage.

## 3.2. Expert-Guided Staged Optimization

GPU kernel optimization involves complex optimization decisions over a vast design space, making end-to-end optimization difficult to control. To address this challenge, we introduce expert optimization workflows to decompose GPU kernel generation into two hierarchical stages: *algorithmic structure design* and *hardware-specific tuning*. Algorithmic structure design determines the computational structure and dataflow organization of the kernel, fundamentally establishing the upper bound of achievable performance. Hardware-specific tuning for the target device focuses on three sub-stages: parallel mapping, tensor tiling, and memory optimization, directly affecting runtime performance. This hierarchical decomposition structures the optimization space and enables progressive, stage-wise refinement.

### 3.2.1. ALGORITHMIC STRUCTURE DESIGN

Algorithmic structure design establishes a high-quality algorithmic foundation that determines the performance upper bound. A key challenge is that the optimal algorithmic approach for a given operator is often non-obvious, and LLM-generated initial implementations frequently contain structural inefficiencies. We address this challenge by combining two complementary strategies: *multi-seed search*, which explores diverse algorithmic paradigms, and *algorithmic refinement*, which optimizes each paradigm to eliminate inefficiencies.

**Multi-Seed Search.** For a given operator, fundamentally different algorithmic paradigms may exist. For example, convolution can be implemented via a direct nested-loop computation or through an `im2col`-based matrix multiplication formulation. Since our structured workflow constrains the optimization space at each subsequent stage, the initial choice of algorithmic paradigm has a significant impact on final performance. Accordingly, as shown in Figure 1, we generate a small set of initial kernel seeds with distinct algorithmic structures to provide diverse starting points. After applying algorithmic refinement to each seed, a lightweight performance filter evaluates the refined seeds and retains only the best-performing candidate for subsequent stages.

**Algorithmic Refinement.** While multi-seed search provides diverse algorithmic directions, each seed's initial implementation typically contains structural inefficiencies. Algorithmic refinement applies expert-guided, semantics-preserving transformations to optimize the computational

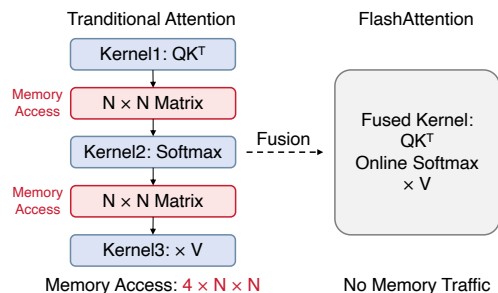

*Figure 2.* **Algorithmic refinement** for the Attention operator. FlashAttention eliminates $4N^2$ memory accesses via kernel fusion.

structure within each seed. Given a seed kernel, the LLM analyzes its operator composition and dataflow structure, and then performs structural optimizations, such as operator fusion, algorithm reformulation, or dataflow reorganization. These refinements reduce redundant computation and unnecessary memory accesses at the algorithmic level.

Figure 2 illustrates the importance of algorithmic refinement using the Attention operator as a representative example. Traditional implementations decompose Attention into multiple independent kernels that compute $QK^T$, apply softmax normalization, and multiply by $V$. For a sequence of length $N$, this approach materializes the $N \times N$ attention matrix in global memory, incurring $4N^2$ additional memory accesses. FlashAttention restructures this computation by introducing an online softmax mechanism that maintains normalization statistics during traversal of $K$ and $V$. This design fuses attention score computation, normalization, and weighted accumulation into a single kernel, eliminating the memory bottleneck.

Overall, the combination of multi-seed search and algorithmic refinement balances coarse-grained paradigm exploration with fine-grained structural optimization, establishing a strong algorithmic foundation for subsequent hardware-specific tuning. We leave the prompt details in Appendix E.

### 3.2.2. HARDWARE-SPECIFIC TUNING

After establishing the high-level algorithmic structure, we perform fine-grained hardware-specific tuning through three progressive stages: *parallel mapping*, which determines the parallel strategy to fully utilize GPU cores; *tensor tiling*, which selects tile-level granularity to maximize on-chip data reuse; and *memory optimization*, which optimizes data access patterns and execution pipelines. Each stage constrains the optimization space of subsequent stages. Parallel mapping fixes the global parallel structure; tensor tiling operates within each GPU execution grid under this structure; and memory optimization further refines execution under fixed parallelization and tiling. By decomposing the joint optimization problem into a sequence of constrained sub-

problems, we provide the LLM with well-scoped objectives at each stage.

**Parallel Mapping.** The parallel mapping stage determines how computational tasks are distributed across the GPU execution grid. The objective is to identify parallelizable dimensions of the operator (e.g., batch, head, or expert dimensions) and map them to the GPU execution grid to fully utilize parallelism. Formally, we define parallel mapping as a dimension-to-grid mapping:

$$(d_1, d_2, \ldots, d_n) \rightarrow (G_1, G_2, \ldots, G_n), \quad (1)$$

where $d_i$ denotes the size of the $i$-th operator dimension and $G_i$ denotes the size of the $i$-th grid dimension.

**Tensor Tiling.** After determining the global parallel structure, the tensor tiling stage focuses on determining the size of tensor tiles processed within each execution grid. We formalize tensor tiling as a dimension-to-tile mapping:

$$(d_1, d_2, \ldots, d_n) \rightarrow (B_1, B_2, \ldots, B_n), \quad (2)$$

where $B_i$ denotes the corresponding tile size. The relationship between parallel mapping and tiling is given by $G_i = \lceil d_i/B_i \rceil$.

Tensor tiling balances computation efficiency and memory behavior. Larger tiles improve on-chip data reuse but incur higher register and shared memory consumption, potentially limiting parallelism. In contrast, smaller tiles reduce per-instance resource usage but often fail to exploit data locality, leading to increased memory access overhead. As a result, tile size selection is inherently a constraint-aware design problem: the choice of $(B_1, \ldots, B_n)$ must respect hardware resource limits while achieving high computational throughput. In EGG, the LLM is guided to propose a small set of tiling candidates, from which the final configuration is selected based on runtime performance measurements.

**Memory Optimization.** Under fixed parallelization and tiling, the stage-specific prompt directs the agent to optimize memory access patterns by organizing global memory accesses into coalesced loads. It also improves software pipelining by adjusting multi-buffering depth to overlap data movement with computation and hide memory access latency.

### 3.3. Stage-Aware Multi-Agent Collaboration

After decomposing the optimization process into distinct stages, we introduce a stage-aware multi-agent collaboration mechanism to perform optimization within each stage. A single agent responsible for code generation, performance analysis, and debugging across all stages often suffers from *objective drift*, where accumulated context obscures the current optimization goal and leads to redundant revisions or regression of prior optimizations. To mitigate this issue,

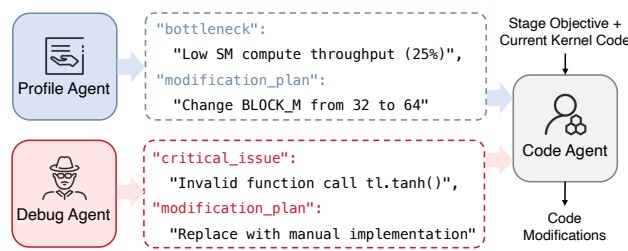

*Figure 3.* Example of intra-stage multi-agent information exchange. The profile and debug agents pass feedback to the code agent via structured JSON outputs.

EGG adopts a collaborative multi-agent design with structured context management, enabling cumulative and stable optimization throughout the staged workflow.

#### 3.3.1. MULTI-AGENT DESIGN

The multi-agent system decomposes each optimization stage into functionally distinct roles. As shown in Figure 1, three specialized agents form a closed-loop collaboration around a unified optimization objective:

- **Profile Agent:** analyzes runtime profiling metrics (e.g., NVIDIA Nsight Compute reports) together with the current kernel code and stage objective, identifies the primary performance bottleneck (e.g., compute-bound, memory-bound), and proposes targeted modification plans to the code agent.

- **Code Agent:** generates the revised kernel according to the modification plan from the profile or debug agent, given the current kernel code and stage objective. Based on runtime results, this kernel is dispatched either to the debug agent for further repair or to the profile agent in the next stage for optimization.

- **Debug Agent:** diagnoses compilation errors, runtime exceptions, or numerical inconsistencies when failures occur, based on error logs and the current kernel code, and outputs targeted fixes to the code agent.

#### 3.3.2. STRUCTURED CONTEXT MANAGEMENT

Context management governs how optimization context is organized, filtered, and shared to ensure effective collaboration among agents. We introduce two complementary mechanisms: *inter-stage context propagation*, which manages context flow across stages to support cumulative optimization, and *intra-stage information exchange*, which coordinates agent interactions within each stage to ensure stable optimization behavior.

**Inter-Stage Context Propagation.** At stage boundaries, we filter and reorganize context to avoid cross-stage interference. When transitioning from stage $t$ to $t + 1$, the system

*Table 1.* Performance comparison on KernelBench. We report success rate, $\text{Fast}_1$ rate, and mean speedup over PyTorch Eager across three difficulty levels.

| Method | Level 1 | | | Level 2 | | | Level 3 | | |
|---|---|---|---|---|---|---|---|---|---|
| | Success(%) | $\text{Fast}_1$(%) | Speedup | Success | $\text{Fast}_1$ | Speedup | Success | $\text{Fast}_1$ | Speedup |
| Torch Compile | 100% | 72% | 1.09× | 100% | 84% | 1.38× | 100% | 92% | 1.36× |
| Deepseek V3.2 | 34% | 11% | 0.99× | 40% | 17% | 0.88× | 36% | 14% | 0.84× |
| ChatGPT 5.1 | 60% | 18% | 0.90× | 60% | 32% | 1.13× | 66% | 24% | 0.91× |
| AutoTriton | 36% | 9% | 1.20× | 55% | 22% | 0.96× | 56% | 26% | 0.83× |
| CudaForge | 100% | 56% | 1.43× | 100% | 90% | 2.00× | 100% | 72% | 1.30× |
| **Ours** | **100%** | **72%** | **1.83×** | **100%** | **100%** | **2.73×** | **100%** | **94%** | **1.52×** |

retains only finalized decisions and kernel implementations from stage $t$, discards intermediate exploratory outputs, and constructs a new context view that explicitly defines the optimization objective for stage $t + 1$. This mechanism establishes a cumulative optimization trajectory without regressing prior improvements.

**Intra-Stage Information Exchange.** Intra-stage information exchange coordinates efficient collaboration among agents through structured JSON interfaces that compress and isolate context. The profile agent provides the bottleneck analysis and a modification plan. When failures occur, the debug agent reports critical issues and the corresponding required fixes. The code agent consumes structured feedback along with the current kernel code and the stage-specific objective to generate code modifications. Figure 3 illustrates this context flow with an example. This structured interface design enables tight collaboration loops across different agents.

Overall, combining multi-agent collaboration with structured context flow management enables cumulative optimization across stages while maintaining stable, focused exploration within each stage.

## 4. Experiments

In this section, we comprehensively evaluate **EGG** through systematic experiments on Triton kernel generation to analyze its effectiveness, robustness, and optimization behavior.

### 4.1. Experimental Setup

**Hardware Platforms.** We show the results performing on NVIDIA GeForce RTX 4090 with 24 GB GDDR6X memory, Ada Lovelace architecture, 128 SMs, and 16,384 CUDA cores; To validate generality across different hardware, we report additional results on RTX 5090, H20, and RTX PRO 6000 GPUs in Appendix A.

**Software.** All experiments are conducted using CUDA 13.0,

PyTorch 2.9.1 and Triton 3.5.1. LLM inference is primarily supported by GPT-5.1. Claude Opus 4.5 results are reported in the Appendix A.

**Benchmark.** We adopt KernelBench (Ouyang et al., 2025) as the primary evaluation benchmark, which consists of 250 kernel tasks spanning three difficulty levels: basic operators, fused operators, and complete networks. Details are reported in the Appendix C.

**Baselines.** We compare EGG against the following baselines: 1) PyTorch Eager, the default execution mode invoking vendor-optimized libraries (e.g., cuBLAS, cuDNN); 2) Torch Compile (Ansel et al., 2024) (default mode), PyTorch's graph compilation framework that applies operator fusion over pre-built kernel libraries; 3) ChatGPT-5.1 (OpenAI, 2025) and 4) DeepSeek-V3.2 (Liu et al., 2025), state-of-the-art general-purpose LLMs; 5) AutoTriton (Li et al., 2025c), a RL-fine-tuned LLM for Triton kernel generation; and 6) CudaForge (Zhang et al., 2025b), a multi-agent framework with performance-feedback-driven iterative refinement. For a fair comparison, we re-evaluate CudaForge under the same setup as EGG. We provide additional comparisons with compiler-based baselines, including Torch Compile in max-autotune mode and TVM Relax (Feng et al., 2023), in the Appendix B.

**Metrics.** Following prior work (Ouyang et al., 2025; Li et al., 2025b; Baronio et al., 2026) and the standard evaluation protocol of KernelBench, we evaluate all methods using three metrics: 1) *Success Rate*: the fraction of tasks that successfully compile and pass correctness verification; 2) $\text{Fast}_1$ *Rate*: the fraction of tasks for which the generated kernels are correct and outperform PyTorch Eager (speedup $> 1.0×$); 3) *Speedup*: the average execution-time improvement over PyTorch Eager, computed over correct kernels.

### 4.2. Overall Performance

Table 1 presents a comprehensive comparison across the three difficulty levels of KernelBench. Overall, EGG consis-

*Table 2.* Ablation study on multi-seed search, algorithmic refinement (Algo Refine), hardware-specific tuning (HW Tune), and multi-agent collaboration.

| Multi-Seed | Algo Refine | HW Tune | Multi-Agent | $Fast_1$ | Speedup |
|:---:|:---:|:---:|:---:|:---:|:---:|
| ✓ | ✓ | ✓ | ✓ | **87.6%** | **2.13×** |
| ✓ | ✓ | ✓ |  | 74.0% | 1.67× |
|  | ✓ | ✓ | ✓ | 78.4% | 1.84× |
|  | ✓ |  | ✓ | 60.8% | 1.52× |
|  |  | ✓ | ✓ | 72.0% | 1.46× |

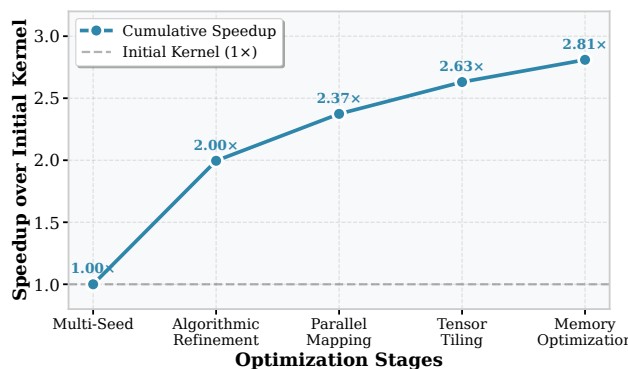

*Figure 4.* Average cumulative speedup across four expert-guided optimization stages.

tently outperforms all baselines as task complexity increases. It achieves a 100% success rate while delivering substantial performance improvements, with an average speedup of 2.13× over PyTorch Eager and 1.60× over the auto-compilation baseline Torch Compile.

**Comparison with PyTorch Eager.** On Level 1 basic operators, EGG achieves a 1.83× speedup with a 72% $Fast_1$ rate and a 100% success rate. As task complexity increases to Level 2 fused operators, the advantage of EGG becomes more pronounced: EGG achieves a 2.73× speedup with a 100% $Fast_1$ rate, where every generated kernel outperforms PyTorch Eager. This result highlights the effectiveness of expert-guided staged optimization for handling fused operators. On the most challenging Level 3 end-to-end models, EGG maintains a 100% success rate while achieving a 1.52× speedup and a 94% $Fast_1$ rate, demonstrating robust performance under complex optimization scenarios.

**Comparison with Other Baselines.** Torch Compile, which leverages graph-level optimizations and pre-built kernel libraries, achieves 1.09–1.38× speedup. However, its reliance on existing kernel implementations limits its effectiveness on novel or highly fused operator patterns. In contrast, EGG synthesizes custom kernels tailored to specific workload patterns, delivering further performance improvements across all difficulty levels.

General-purpose LLMs (DeepSeek and ChatGPT) achieve only 11-32% $Fast_1$ rates with 34–66% success rates due to the lack of domain expertise. The RL-based AutoTriton attempts to address this limitation by fine-tuning the model with execution-time reward signals. However, due to the scarcity of high-quality kernels, effective exploration of the optimization space remains challenging.

CudaForge employs multi-agent iterative refinement driven by hardware feedback and achieves 1.30-2.00× speedup. However, without expert guidance on the optimization direction, it relies on trial-and-error exploration, limiting its effectiveness on complex tasks (e.g., 72% $Fast_1$ rate on Level 3). In contrast, EGG guides kernel generation with expert optimization principles, enabling higher performance.

**Cost Efficiency.** Under the same GPT-5.1 and RTX 4090 setup, EGG completes kernel generation for a single task in approximately 20 minutes, consuming around 50,000 output tokens per kernel. In comparison, CudaForge takes approximately 30 minutes and consumes around 110,000 output tokens per kernel under the same setup. This result demonstrates that decomposing kernel optimization into stage-wise objectives improves not only final kernel quality but also search efficiency.

### 4.3. Ablation Study

As shown in Table 2, we systematically analyze the contribution of individual components in **EGG** through a set of ablation studies.

**Effect of Multi-Seed Search.** Introducing multi-seed search improves the average speedup from 1.84× to 2.13× and increases the $Fast_1$ rate from 78.4% to 87.6%. The gains vary with task complexity: Level 2 and Level 3 achieve additional speedups of 1.7× and 2.0×, respectively, whereas Level 1 shows only a modest improvement of 1.09×. This trend arises because complex tasks expose larger algorithmic design spaces, allowing broader exploration, while simple operators tend to converge to similar implementations. Overall, multi-seed search mitigates early convergence to suboptimal structures, with benefits increasing as task complexity grows.

**Stage-Specific Contributions.** Enabling only the algorithmic refinement stage yields a 1.52× speedup but achieves a moderate $Fast_1$ rate of 60.8%. This suggests that while algorithmic refinement can unlock high performance potential, it does not consistently deliver speedups across kernels without hardware-specific optimization. Conversely, enabling only hardware-specific tuning achieves a 1.46× speedup with a higher $Fast_1$ rate of 72%, providing more stable acceleration across tasks but limited peak performance due to suboptimal initial algorithmic choices. Combining both components achieves the best overall performance, demon-

*Table 3.* Optimization trajectory for grouped 3D transposed convolution: latency and speedup relative to PyTorch Eager.

| Stage | Latency | Speedup |
|---|---|---|
| PyTorch Eager | 20.78ms | – |
| Seed 1 | 29.62ms | 0.70× |
| Seed 2 | 76.04ms | 0.27× |
| Algorithmic Refinement (S1) | 19.41ms | 1.07× |
| Algorithmic Refinement (S2) | 12.38ms | 1.68× |
| Parallel Mapping | 10.78ms | 1.93× |
| Tensor Tiling | 9.20ms | 2.26× |

*Table 4.* Execution time comparison for representative real-world Triton workloads.

| Task | TritonBench | Ours |
|---|---|---|
| Flash Attention | 0.046 ms | **0.037 ms** |
| RoPE Embedding | 0.072 ms | **0.044 ms** |
| INT8 Dequant MatMul | 0.067 ms | **0.062 ms** |

strating their complementary roles: algorithmic refinement establishes the performance ceiling, while hardware-specific tuning ensures reliable realization of that potential.

**Multi-Agent Contributions.** Using only a single code agent limits the system's ability to identify performance bottlenecks and correctness issues, leading to degraded performance and lower success rates. This result highlights the importance of stage-aware multi-agent collaboration, where coordination among specialized agents enables consistent and stable performance improvements.

**Cumulative Effects Across Stages.** Figure 4 illustrates the average cumulative performance improvement across four expert-guided optimization stages. The algorithmic refinement stage achieves an initial $2.0\times$ speedup over the raw seed implementation, effectively establishing a strong performance baseline. Subsequent hardware-specific tuning stages contribute an additional $1.7\times$ improvement through parallel mapping ($1.33\times$), tensor tiling ($1.08\times$), and memory optimization ($1.13\times$). These results indicate that EGG achieves high performance through the accumulation of improvements across different expert-guided stages.

### 4.4. Case Study: 3D Transposed Convolution

We demonstrate the optimization workflow of **EGG** using a grouped 3D transposed convolution operator with batch size 16, 32 channels, kernel size $(3, 5, 7)$, stride $(2, 2, 2)$, and 4 groups. As shown in Table 3, the PyTorch Eager baseline executes this operator in 20.78 ms.

**Algorithmic Structure Design.** *Multi-Seed Search.* The multi-seed search generates two initial implementations with distinct algorithmic structures. Seed 1 adopts a backward-mapping strategy, achieving a latency of 29.62 ms ($0.7\times$ speedup). Seed 2 employs nested iteration over input elements and kernel dimensions, resulting in a significantly higher latency of 76.04 ms ($0.27\times$ speedup). Both initial implementations underperform the PyTorch Eager baseline.

*Algorithmic Refinement.* For Seed 1, the profile agent iden-

tifies that backward mapping requires expensive division and modulo operations when enumerating the $C_I \times K_D \times K_H \times K_W$ combinations, leading to control-flow overhead and warp divergence. The code agent restructures the kernel to a forward-mapping formulation with in-block accumulation, reducing the latency to 19.41 ms ($1.07\times$). For Seed 2, the profile agent diagnoses low arithmetic intensity as the primary bottleneck. By transforming the transposed convolution into an `im2col`-based matrix multiplication, the refined kernel reduces latency to 12.38 ms ($1.68\times$). This refined implementation is selected for subsequent hardware-specific tuning.

**Hardware-Specific Tuning.** *Parallel mapping.* NCU profiling reveals low streaming multiprocessor (SM) utilization. Guided by the profile agent, the code agent remaps output pixels, output channels, and groups across three `program_id` axes, improving workload distribution across SMs and reducing latency to 10.78 ms ($1.93\times$).

*Tensor tiling.* Further analysis of register and shared memory usage guides autotuning over `BLOCKM/N/K` configurations. Among $(32, 32, 32)$, $(64, 32, 32)$, and $(32, 64, 32)$, the optimal configuration achieves a latency of 9.20 ms, corresponding to a $2.26\times$ speedup over PyTorch Eager.

Table 3 summarizes the complete optimization trajectory. The algorithmic structure design delivers a $1.68\times$ improvement over PyTorch Eager, where the algorithmic refinement achieves a $6.2\times$ improvement over the worst seed, establishing a strong foundation. Subsequent hardware-specific tuning contributes an additional cumulative $1.35\times$ gain through parallel mapping ($1.15\times$) and tensor tiling ($1.17\times$), achieving a final $2.26\times$ speedup over PyTorch Eager and an $8.4\times$ improvement over the initial seed.

### 4.5. Practical Application Verification

To evaluate practical deployment effectiveness, we assess EGG on representative operators from TritonBench (Li et al., 2025b), as shown in Table 4. These operators are derived from production Triton kernels in GitHub repositories and commonly used in real-world LLM workloads.

We compare our generated kernels against the original handwritten Triton implementations. EGG achieves substantial speedups: $1.24\times$ for Flash Attention, $1.63\times$ for RoPE Em-

bedding, and $1.08\times$ for INT8 Dequant MatMul. These results demonstrate that expert manual tuning does not always achieve optimal performance for complex operators. EGG effectively harnesses LLMs' exploration capabilities to discover implementations that surpass hand-tuned production kernels, validating the practical value of our approach.

## 5. Conclusion

In this work, we propose an expert-guided GPU kernel generation agent framework that decomposes optimization into algorithmic structure design and hardware-specific tuning, guiding LLM decisions with expert optimization workflows. A stage-aware multi-agent collaboration mechanism coordinates code, profile, and debug agents to achieve stable and cumulative improvements. Experiments on KernelBench and real-world workloads demonstrate an average $2.13\times$ speedup over PyTorch, outperforming existing agent-based and RL-based approaches.

## Impact Statement

This paper presents work whose goal is to advance the field of Machine Learning. There are many potential societal consequences of our work, none which we feel must be specifically highlighted here.

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

## Overview

This appendix provides additional experimental evidence and implementation details to complement the main paper. It is organized as follows:

- **Additional Hardware and LLM Results (Section A):** We report EGG's performance on additional GPU platforms, including NVIDIA RTX 5090, H20, and RTX PRO 6000, and evaluate Claude Opus 4.5 to demonstrate model-agnostic effectiveness.

- **Additional Baseline Results (Section B):** We provide additional comparisons with `torch.compile` in `max-autotune` mode and TVM Relax to further validate EGG against compiler-based baselines.

- **Benchmark Details (Section C):** We summarize the task structure and interface conventions of KernelBench, and describe the construction of TritonBench.

- **Reproducibility Details (Section D):** We summarize the fixed interaction budget, stopping criteria, deterministic failure handling, and pseudocode of the EGG optimization pipeline.

- **Prompt Details (Section E):** We provide the full prompts used in our framework, including the seed prompt, stage-specific system prompts, and the profile/code/debug agent prompts used for optimization and repair.

- **Nsight Compute Profiling Metrics (Section F):** We list the NCU metrics collected during profiling, and explain how they guide targeted optimization decisions in our framework.

- **Limitations (Section G):** We list some limitations in our current work, which we treat as our future work for further improvement.

## A. Additional Hardware and LLM Results

### A.1. Overall Performance Across Hardware Platforms

To validate the generality of EGG across different GPU architectures, we conduct additional experiments on NVIDIA GeForce RTX 5090, NVIDIA H20, and NVIDIA RTX PRO 6000 GPUs. The RTX 5090 features the Blackwell architecture with 170 SMs, 21,760 CUDA cores, and 32GB GDDR7 memory. The H20 is a Hopper-based data-center GPU with 96GB HBM3 memory, while the RTX PRO 6000 is a Blackwell-based professional workstation GPU with 96GB GDDR7 memory. Table 5 summarizes the RTX 5090 results across all three KernelBench levels, while Table 6 summarizes the H20 and RTX PRO 6000 results on the Level 2 fused operators. Across these evaluations, EGG maintains high correctness and consistently achieves strong speedups over PyTorch Eager, demonstrating the architectural robustness of our framework beyond the RTX 4090 setting used in the main text.

*Table 5.* Performance summary on RTX 5090 across KernelBench difficulty levels.

| Level | Success Rate (%) | Fast$_1$ Rate (%) | Avg. Speedup ($\times$) |
|---|---|---|---|
| Level 1 (Basic) | 100 | 69 | 2.40 |
| Level 2 (Medium) | 100 | 96 | 2.67 |
| Level 3 (Hard) | 100 | 90 | 1.41 |
| **Overall** | **100** | **84** | **2.31** |

*Table 6.* Performance summary on additional GPU platforms for KernelBench Level 2 fused operators.

| Hardware | Success Rate (%) | Fast$_1$ Rate (%) | Avg. Speedup ($\times$) |
|---|---|---|---|
| NVIDIA H20 | 100 | 88 | 3.23 |
| NVIDIA RTX PRO 6000 | 100 | 93 | 2.95 |

### A.2. Results with Opus 4.5

To assess the model-agnostic nature of our agent framework, we conduct additional experiments using Claude Opus 4.5 on the medium-difficulty Level 2 tasks (fused operators) of KernelBench. As summarized in Table 7, our framework achieves consistently strong performance. These results indicate that the proposed agent framework generalizes well across different LLMs.

*Table 7.* Performance results using Opus 4.5 on KernelBench.

| Level | Success Rate (%) | Fast$_1$ Rate (%) | Avg. Speedup ($\times$) |
|-------|------------------|-------------------|-------------------------|
| Level 2 | 100 | 99 | 2.95 |

## B. Additional Baseline Results

This section provides additional baseline comparisons to further validate the effectiveness of EGG. We include `torch.compile` in `max-autotune` mode and TVM Relax as compiler-based baselines, complementing the main results.

### B.1. Comparison with `torch.compile` in `max-autotune` mode

In the main experiments, `torch.compile` is evaluated under its `default` mode. We further evaluate `torch.compile` with `max-autotune`, which enables more aggressive autotuning and typically provides stronger performance. As shown in Table 8, EGG still achieves clear performance gains across all KernelBench levels.

*Table 8.* Comparison with `torch.compile` in `max-autotune` mode on KernelBench.

| Level | `torch.compile`(`max-autotune`) | | EGG | |
|-------|-----------|------------|-----------|------------|
| | **Fast1 (%)** | **Speedup ($\times$)** | **Fast1 (%)** | **Speedup ($\times$)** |
| Level 1 | 66 | 1.04 | 72 | 1.83 |
| Level 2 | 87 | 1.51 | 100 | 2.73 |
| Level 3 | 84 | 1.48 | 94 | 1.52 |

### B.2. Comparison with TVM Relax

We also compare EGG with TVM Relax, a traditional machine learning compiler baseline. TVM Relax applies compiler-defined graph transformations and schedule optimizations to improve deep learning workloads. We evaluate TVM Relax on 25 representative KernelBench tasks selected from the three difficulty levels. EGG achieves a 1.92$\times$ geometric mean speedup over PyTorch Eager and a 1.56$\times$ speedup over TVM Relax on the same task subset. The performance gap is smaller on Level 3, where TVM Relax benefits from graph-level optimization. In contrast, EGG shows clearer advantages on Level 1 and Level 2 tasks, where its gains mainly come from expert-guided computation restructuring and workload-specific kernel synthesis.

*Table 9.* Comparison with TVM Relax on 25 representative KernelBench tasks.

| Level | #Tasks | TVM Relax | EGG | EGG / TVM |
|-------|--------|-----------|-----|-----------|
| Level 1 | 8 | 1.18$\times$ | 2.26$\times$ | 1.91$\times$ |
| Level 2 | 11 | 1.09$\times$ | 1.94$\times$ | 1.78$\times$ |
| Level 3 | 6 | 1.51$\times$ | 1.55$\times$ | 1.03$\times$ |
| **Overall** | **25** | **1.23$\times$** | **1.92$\times$** | **1.56$\times$** |

## C. Benchmark Details

This section provides detailed specifications of the benchmarks used in our evaluation, including representative task examples with their PyTorch reference implementations.

### C.1. KernelBench

KernelBench is a recently proposed benchmark specifically designed for evaluating LLM-based GPU kernel optimization frameworks. The benchmark comprises 250 carefully curated tasks organized into three difficulty tiers:

- **Level 1 (Basic):** 100 fundamental operators including arithmetic operations (matrix multiplication, convolution), element-wise operations (ReLU, GELU, sigmoid), and basic reductions (softmax, layer normalization).

- **Level 2 (Medium):** 100 medium-difficulty tasks that fuse multiple primitive operations, such as GEMM+ReLU+Add and Conv2D+BatchNorm+ReLU.

- **Level 3 (Hard):** 50 challenging tasks implementing complete neural network architectures and complex computational patterns, including AlexNet, ResNet and Vision Transformer.

All tasks adhere to a standardized interface design that facilitates automated evaluation. Each task provides a reference PyTorch implementation following a consistent structure: a `Model` class inheriting from `nn.Module` with a `forward()` method defining the computation, a `get_inputs()` function generating runtime inputs, and a `get_init_inputs()` function providing model initialization parameters. This uniform interface enables reliable correctness verification through numerical comparison and consistent performance benchmarking across different implementations.

Below we present one representative example from each difficulty level to illustrate the benchmark's task structure and interface conventions.

**Level 1: Square Matrix Multiplication**

```
import torch
import torch.nn as nn

class Model(nn.Module):
    """
    Simple model that performs a single square matrix multiplication (C = A * B)
    """
    def __init__(self):
        super(Model, self).__init__()

    def forward(self, A: torch.Tensor, B: torch.Tensor) -> torch.Tensor:
        """
        Performs the matrix multiplication.

        Args:
            A (torch.Tensor): Input matrix A of shape (N, N).
            B (torch.Tensor): Input matrix B of shape (N, N).

        Returns:
            torch.Tensor: Output matrix C of shape (N, N).
        """
        return torch.matmul(A, B)

N = 2048 * 2

def get_inputs():
    A = torch.rand(N, N)
    B = torch.rand(N, N)
    return [A, B]

def get_init_inputs():
```

```
        return []  # No special initialization inputs needed
```

### Level 2: Conv2D + ReLU + BiasAdd

```python
import torch
import torch.nn as nn

class Model(nn.Module):
    """
    Simple model that performs a convolution, applies ReLU, and adds a bias term.
    """
    def __init__(self, in_channels, out_channels, kernel_size, bias_shape):
        super(Model, self).__init__()
        self.conv = nn.Conv2d(in_channels, out_channels, kernel_size)
        self.bias = nn.Parameter(torch.randn(bias_shape))

    def forward(self, x):
        x = self.conv(x)
        x = torch.relu(x)
        x = x + self.bias
        return x

batch_size = 128
in_channels  = 64
out_channels = 128
height = width = 128
kernel_size = 3
bias_shape = (out_channels, 1, 1)

def get_inputs():
    return [torch.rand(batch_size, in_channels, height, width)]

def get_init_inputs():
    return [in_channels, out_channels, kernel_size, bias_shape]
```

### Level 3: Multi-Layer Perceptron (MLP)

```python
import torch
import torch.nn as nn
import torch.nn.functional as F

class Model(nn.Module):
    def __init__(self, input_size, layer_sizes, output_size):
        """
        :param input_size: The number of input features
        :param layer_sizes: A list of ints containing the sizes of each hidden layer
        :param output_size: The number of output features
        """
        super(Model, self).__init__()

        layers = []
        current_input_size = input_size

        for layer_size in layer_sizes:
            layers.append(nn.Linear(current_input_size, layer_size))
            layers.append(nn.ReLU())
            current_input_size = layer_size

        layers.append(nn.Linear(current_input_size, output_size))

        self.network = nn.Sequential(*layers)
```

```
    def forward(self, x):
        """
        :param x: The input tensor, shape (batch_size, input_size)
        :return: The output tensor, shape (batch_size, output_size)
        """
        return self.network(x)

# Test code
batch_size = 128
input_size = 16384
layer_sizes = [16384, 16384]
output_size = 8192

def get_inputs():
    return [torch.rand(batch_size, input_size)]

def get_init_inputs():
    return [input_size, layer_sizes, output_size]
```

### C.2. TritonBench

TritonBench features 184 real-world operators collected from GitHub repositories (>100 stars), providing a diverse collection of Triton kernels spanning various computational patterns and optimization levels. The dataset includes kernels for matrix operations, convolution operations, attention mechanisms, and custom computational kernels, each with corresponding unit tests. It provides natural language descriptions as input, we convert them to PyTorch reference implementations to align with practical kernel optimization scenarios.

Below we show an example Rotary Position Embedding (RoPE) operator, presenting both the original Triton implementation from GitHub (https://github.com/turbo-llm/turbo-alignment) and our converted PyTorch reference.

**Original Triton Implementation:**

```
import torch
import triton
import triton.language as tl

@triton.jit
def _triton_rope(
    q_ptr,
    q_row_stride,
    k_ptr,
    k_row_stride,
    cos,
    cos_row_stride,
    sin,
    sin_row_stride,
    sl,
    bs: tl.constexpr,
    n_qh: tl.constexpr,
    n_kh: tl.constexpr,
    hd: tl.constexpr,
    pad_n_qh: tl.constexpr,
    pad_n_kh: tl.constexpr,
    pad_hd: tl.constexpr,
    BLOCK_SIZE: tl.constexpr,
    BACKWARD_PASS: tl.constexpr = False,
):
    pid = tl.program_id(0)

    q_ptr = q_ptr + pid * q_row_stride
```

```
    k_ptr = k_ptr + pid * k_row_stride

    cos_row_idx = pid % (sl)
    cos = cos + cos_row_idx * cos_row_stride
    sin = sin + cos_row_idx * sin_row_stride
    cos_offsets = tl.arange(0, pad_hd // 2)
    cos_mask = cos_offsets < hd // 2
    cos_row = tl.load(cos + cos_offsets, mask=cos_mask, other=0)
    sin_row = tl.load(sin + cos_offsets, mask=cos_mask, other=0)

    first_half_q_offsets = tl.arange(0, pad_n_qh)[:, None] * hd + tl.arange(0, pad_hd
    // 2)[None, :]
    first_half_k_offsets = tl.arange(0, pad_n_kh)[:, None] * hd + tl.arange(0, pad_hd
    // 2)[None, :]
    first_q_mask = (tl.arange(0, pad_n_qh)[:, None] < n_qh) & (tl.arange(0, pad_hd //
    2)[None, :] < hd // 2)
    first_k_mask = (tl.arange(0, pad_n_kh)[:, None] < n_kh) & (tl.arange(0, pad_hd //
    2)[None, :] < hd // 2)
    q_tile_1 = tl.load(q_ptr + first_half_q_offsets, mask=first_q_mask,
    other=0).to(sin_row.dtype)
    k_tile_1 = tl.load(k_ptr + first_half_k_offsets, mask=first_k_mask,
    other=0).to(sin_row.dtype)

    second_half_q_offsets = first_half_q_offsets + (hd // 2)
    second_half_k_offsets = first_half_k_offsets + (hd // 2)
    second_q_mask = first_q_mask
    second_k_mask = first_k_mask
    q_tile_2 = tl.load(q_ptr + second_half_q_offsets, mask=second_q_mask,
    other=0).to(sin_row.dtype)
    k_tile_2 = tl.load(k_ptr + second_half_k_offsets, mask=second_k_mask,
    other=0).to(sin_row.dtype)

    if not BACKWARD_PASS:
        new_q_tile_1 = q_tile_1 * cos_row - q_tile_2 * sin_row
        tl.store(q_ptr + first_half_q_offsets, new_q_tile_1, mask=first_q_mask)
        new_q_tile_2 = q_tile_2 * cos_row + q_tile_1 * sin_row
        tl.store(q_ptr + second_half_q_offsets, new_q_tile_2, mask=second_q_mask)

        new_k_tile_1 = k_tile_1 * cos_row - k_tile_2 * sin_row
        tl.store(k_ptr + first_half_k_offsets, new_k_tile_1, mask=first_k_mask)
        new_k_tile_2 = k_tile_2 * cos_row + k_tile_1 * sin_row
        tl.store(k_ptr + second_half_k_offsets, new_k_tile_2, mask=second_k_mask)
    else:
        new_q_tile_1 = q_tile_1 * cos_row + q_tile_2 * sin_row
        tl.store(q_ptr + first_half_q_offsets, new_q_tile_1, mask=first_q_mask)
        new_q_tile_2 = q_tile_2 * cos_row - q_tile_1 * sin_row
        tl.store(q_ptr + second_half_q_offsets, new_q_tile_2, mask=second_q_mask)

        new_k_tile_1 = k_tile_1 * cos_row + k_tile_2 * sin_row
        tl.store(k_ptr + first_half_k_offsets, new_k_tile_1, mask=first_k_mask)
        new_k_tile_2 = k_tile_2 * cos_row - k_tile_1 * sin_row
        tl.store(k_ptr + second_half_k_offsets, new_k_tile_2, mask=second_k_mask)

def rope_forward(q, k, cos, sin):
    q = q.transpose(1, 2)
    k = k.transpose(1, 2)

    batch_size, seq_len, n_q_head, head_dim = q.shape
    n_kv_head = k.shape[2]
    pad_hd = triton.next_power_of_2(head_dim)
    pad_n_q_head = triton.next_power_of_2(n_q_head)
    pad_n_kv_head = triton.next_power_of_2(n_kv_head)
```

```
        BLOCK_SIZE = max(pad_n_q_head, pad_n_kv_head)

    n_row = batch_size * seq_len

    q = q.contiguous()
    k = k.contiguous()
    cos = cos.contiguous()
    sin = sin.contiguous()

    _triton_rope[(n_row,)](
        q,
        q.stride(1),
        k,
        k.stride(1),
        cos,
        cos.stride(-2),
        sin,
        sin.stride(-2),
        seq_len,
        batch_size,
        n_q_head,
        n_kv_head,
        head_dim,
        pad_n_q_head,
        pad_n_kv_head,
        pad_hd,
        BLOCK_SIZE=BLOCK_SIZE,
        BACKWARD_PASS=False,
    )
    return q.transpose(1, 2), k.transpose(1, 2), cos, sin

import torch

def test_rope_forward():
    # Define the test parameters
    batch_size = 2
    seq_len = 4
    n_q_head = 8
    n_kv_head = 8
    head_dim = 16

    # Create random input tensors
    q = torch.randn(batch_size, n_q_head, seq_len, head_dim, dtype=torch.float32,
    device='cuda')
    k = torch.randn(batch_size, n_kv_head, seq_len, head_dim, dtype=torch.float32,
    device='cuda')
    cos = torch.randn(seq_len, head_dim // 2, dtype=torch.float32, device='cuda')
    sin = torch.randn(seq_len, head_dim // 2, dtype=torch.float32, device='cuda')

    # Dictionary to store results for each test case
    results = {}

    # Test case 1: Forward pass
    q_out_1, k_out_1, cos_out_1, sin_out_1 = rope_forward(q, k, cos, sin)
    results['test_case_1'] = (q_out_1, k_out_1, cos_out_1, sin_out_1)

    # Test case 2: Backward pass
    q_out_2, k_out_2, cos_out_2, sin_out_2 = rope_forward(q, k, cos, sin)
    results['test_case_2'] = (q_out_2, k_out_2, cos_out_2, sin_out_2)

    return results

result_gold = test_rope_forward()
```

**Our Converted PyTorch Reference Implementation:**

```python
import torch
import torch.nn as nn

class Model(nn.Module):
    """
    RoPE (Rotary Position Embedding) - PyTorch Reference Implementation

    Rotary Position Embedding applies a rotation to the query and key vectors
    based on their position in the sequence. This allows the model to naturally
    encode relative positions.

    Formula:
        For each position, split the embedding into two halves [x1, x2]
        Apply rotation: [x1*cos - x2*sin, x2*cos + x1*sin]

    Used in: LLaMA, GPT-J, GPT-NeoX, PaLM, and many modern LLMs
    """
    def __init__(self):
        super(Model, self).__init__()

    def forward(self, q, k, cos, sin):
        """
        Apply RoPE to query and key tensors.

        Args:
            q (torch.Tensor): Query tensor of shape (batch, n_heads, seq_len,
    head_dim)
            k (torch.Tensor): Key tensor of shape (batch, n_heads, seq_len, head_dim)
            cos (torch.Tensor): Cosine values of shape (seq_len, head_dim//2)
            sin (torch.Tensor): Sine values of shape (seq_len, head_dim//2)

        Returns:
            Tuple[torch.Tensor, torch.Tensor, torch.Tensor, torch.Tensor]:
                - q_rotated: Rotated query (batch, n_heads, seq_len, head_dim)
                - k_rotated: Rotated key (batch, n_heads, seq_len, head_dim)
                - cos: Cosine values (unchanged)
                - sin: Sine values (unchanged)
        """
        # Transpose to (batch, seq_len, n_heads, head_dim) for easier position-wise
    operation
        q = q.transpose(1, 2)
        k = k.transpose(1, 2)

        batch_size, seq_len, n_heads, head_dim = q.shape
        half_dim = head_dim // 2

        # Split into two halves along head_dim
        q1 = q[..., :half_dim]  # First half
        q2 = q[..., half_dim:]  # Second half
        k1 = k[..., :half_dim]
        k2 = k[..., half_dim:]

        # Reshape cos/sin for broadcasting: (seq_len, head_dim//2) -> (1, seq_len, 1,
    head_dim//2)
        cos = cos[:seq_len, :].unsqueeze(0).unsqueeze(2)
        sin = sin[:seq_len, :].unsqueeze(0).unsqueeze(2)

        # Apply rotation transformation
        # RoPE formula: rotate_half([x1, x2]) = [x1*cos - x2*sin, x2*cos + x1*sin]
        q_rotated = torch.cat([
            q1 * cos - q2 * sin,  # New first half
```

```
            q2 * cos + q1 * sin   # New second half
        ], dim=-1)

        k_rotated = torch.cat([
            k1 * cos - k2 * sin,
            k2 * cos + k1 * sin
        ], dim=-1)

        # Transpose back to (batch, n_heads, seq_len, head_dim)
        q_rotated = q_rotated.transpose(1, 2)
        k_rotated = k_rotated.transpose(1, 2)

        # Return cos/sin as well to match Triton interface
        return q_rotated, k_rotated, cos.squeeze(0).squeeze(1),
    sin.squeeze(0).squeeze(1)

BATCH_SIZE = 2
N_HEADS = 8
SEQ_LEN = 4
HEAD_DIM = 16

def get_inputs():
    """
    Generate test inputs for RoPE.

    Returns:
        List containing [q, k, cos, sin]:
            - q: Query tensor (batch, n_heads, seq_len, head_dim)
            - k: Key tensor (batch, n_heads, seq_len, head_dim)
            - cos: Cosine values (seq_len, head_dim//2)
            - sin: Sine values (seq_len, head_dim//2)
    """
    q = torch.randn(BATCH_SIZE, N_HEADS, SEQ_LEN, HEAD_DIM, dtype=torch.float32)
    k = torch.randn(BATCH_SIZE, N_HEADS, SEQ_LEN, HEAD_DIM, dtype=torch.float32)
    cos = torch.randn(SEQ_LEN, HEAD_DIM // 2, dtype=torch.float32)
    sin = torch.randn(SEQ_LEN, HEAD_DIM // 2, dtype=torch.float32)
    return [q, k, cos, sin]

def get_init_inputs():
    """
    Get initialization parameters for Model.

    Returns:
        Empty list (no initialization parameters needed)
    """
    return []
```

## D. Reproducibility Details

EGG follows the optimization pipeline in Algorithm 1. Validation and benchmarking run in isolated spawned subprocesses to avoid CUDA-context contamination. For each task, the code agent generates two seed kernels; each seed is validated, repaired by the debug agent up to three times if needed, and benchmarked once valid. If no seed becomes valid, the task terminates early. For each valid seed, EGG collects NCU metrics and invokes the profile agent for one algorithmic analysis. If the analysis returns "not worth optimizing", the seed is kept unchanged; otherwise, the code agent performs one algorithmic refinement. EGG then selects the best valid candidate and applies three hardware-specific tuning stages: parallel mapping, tensor tiling, and memory optimization. In each hardware-specific tuning stage, the profile agent uses NCU metrics to produce a stage-specific diagnosis before the code agent modifies the kernel. Both algorithmic refinement and hardware-specific tuning follow the same evaluation rule: the generated candidate is validated, repaired once by the debug agent if needed, and benchmarked once valid. A candidate replaces the current best only if it is valid and faster; otherwise, EGG retains the previous best.

**Algorithm 1** EGG optimization pipeline.

**Require:** Task specification $T$
**Ensure:** Best validated candidate, or failure
1:  $\mathcal{S} \leftarrow \emptyset$
2:  **for** $i = 1$ to 2 **do**
3:     $c \leftarrow$ CODEAGENT.GENERATESEED$(T)$
4:     **for** $r = 0$ to 3 **do**
5:         **if** VALIDATE$(c)$ succeeds **then**
6:             $p \leftarrow$ BENCHMARK$(c)$
7:             $\mathcal{S} \leftarrow \mathcal{S} \cup \{(c, p)\}$
8:             **break**
9:         **else if** $r < 3$ **then**
10:            $c \leftarrow$ DEBUGAGENT$(T, c, \text{failure log})$
11:         **end if**
12:     **end for**
13: **end for**
14: **if** $\mathcal{S} = \emptyset$ **then**
15:     **return** failure
16: **end if**
17: $\mathcal{C} \leftarrow \mathcal{S}$
18: **for all** $(c, p) \in \mathcal{S}$ **do**
19:     $m \leftarrow$ PROFILENCU$(c)$
20:     $a \leftarrow$ PROFILEAGENT$(T, c, p, m)$
21:     **if** $a$ is "not worth optimizing" **then**
22:         **continue**
23:     **end if**
24:     $c' \leftarrow$ CODEAGENT.ALGORITHMICREFINEMENT$(T, c, a)$
25:     **if** VALIDATE$(c')$ fails **then**
26:         $c' \leftarrow$ DEBUGAGENT$(T, c', \text{failure log})$
27:     **end if**
28:     **if** VALIDATE$(c')$ succeeds **then**
29:         $p' \leftarrow$ BENCHMARK$(c')$
30:         $\mathcal{C} \leftarrow \mathcal{C} \cup \{(c', p')\}$
31:     **end if**
32: **end for**
33: $(best, p_{best}) \leftarrow$ best-performing candidate in $\mathcal{C}$
34: **for all** $s \in \{$PARALLELMAPPING, TENSORTILING, MEMORYOPTIMIZATION$\}$ **do**
35:     $m \leftarrow$ PROFILENCU$(best)$
36:     $h \leftarrow$ PROFILEAGENT$(s, T, best, p_{best}, m)$
37:     $c \leftarrow$ CODEAGENT.HARDWARETUNING$(s, T, best, h)$
38:     **if** VALIDATE$(c)$ fails **then**
39:         $c \leftarrow$ DEBUGAGENT$(T, c, \text{failure log})$
40:     **end if**
41:     **if** VALIDATE$(c)$ succeeds **then**
42:         $p \leftarrow$ BENCHMARK$(c)$
43:         **if** $p > p_{best}$ **then**
44:             $(best, p_{best}) \leftarrow (c, p)$
45:         **end if**
46:     **end if**
47: **end for**
48: **return** $best$

Failure handling is deterministic. Compilation failures, runtime failures, and accuracy failures are sent to the debug agent with the corresponding failure log. NCU profiling failure is non-fatal: EGG continues with the current best validated kernel and an empty profiling block. Malformed or truncated LLM outputs are retried once; if the retry also fails, the output is discarded and EGG retains the current best candidate.

## E. Prompt Details

This section provides the complete prompts used in our multi-agent framework. These prompts encode domain-specific optimization principles and guide agents through different stages of kernel generation.

### E.1. Seed Prompt

The seed prompt is used during the multi-seed search stage to generate initial Triton kernel implementations. It receives the target PyTorch operator and few-shot examples as input. The prompt emphasizes strict syntactic constraints and common pitfalls to ensure high initial correctness. It outputs initial Triton kernel implementations with diverse algorithmic structures.

```
Write high-performance Triton kernels to replace PyTorch operators.
Generate the FASTEST kernel while maintaining correctness.

## CRITICAL -- These cause 60%+ of failures:
1. EVERY kernel function MUST have `@triton.jit` decorator -- MANDATORY
2. Grid size MUST be > 0: use `triton.cdiv(N, BLOCK)` or `max(1, N // BLOCK)`
3. BLOCK sizes MUST be power-of-2 constexpr: 16, 32, 64, 128, 256
4. `tl.program_id(axis)` only supports axis = 0, 1, 2 (max 3D grid)

## Triton Syntax Rules:
- For matmul/conv/linear ops, prefer `tl.dot(a, b, allow_tf32=True)` over
    element-wise multiply-add
- No `continue`, `break`, `return` inside loops -- use masking instead
- No tensor indexing with loop vars: `x[:, i]` or `x[i, :]` is INVALID
- No tuple unpacking inside kernel: `a, b = tl.load(...)` is INVALID
- No nested functions inside @triton.jit
- No Python control flow on tl.tensor or BLOCK_* values
- No dynamic `tl.reshape()` or view operations

## Missing Triton Functions (implement manually):
- tl.tanh -- `(tl.exp(2*x) - 1) / (tl.exp(2*x) + 1)`
- tl.sigmoid -- `1 / (1 + tl.exp(-x))`
- tl.gelu, tl.silu, tl.softmax, tl.mish -- implement from definition

## Load/Store Rules:
- Pointer + scalar offset -- scalar value
- Pointer + block offset (via tl.arange) -- block of values
- mask shape MUST match data shape exactly

## Output Format (STRICT):
1. Imports: `import torch, torch.nn as nn, triton, triton.language as tl` (and math
    if needed)
2. `@triton.jit` kernel(s) -- MUST have this decorator
3. Wrapper function with grid calculation
4. `class ModelNew(nn.Module)` -- REQUIRED

Do NOT include: testing code, `if __name__ == "__main__"`, get_inputs, get_init_inputs

Example PyTorch:
'''
$few_base
'''

Example Triton:
'''
```

```
$few_new
'''

Target:
```python
$kernel_src
```
"""
```

## E.2. Stage System Prompts for Hardware-Specific Tuning

During hardware-specific tuning, each optimization stage uses a specialized system prompt that defines the stage's focus, relevant metrics, and optimization rules. These prompts constrain agent decisions to stage-specific objectives, preventing cross-stage interference.

### E.2.1. GRID AND PARALLEL MAPPING

This stage focuses on determining the optimal mapping between operator dimensions and GPU grid dimensions. The prompt guides the agent to prioritize batch/head/expert parallelism before reducing block sizes, and ensures grid configurations remain within hardware limits (maximum 3 dimensions).

```
Focus: Grid layout & parallelism.

Metrics:
- sm__throughput.avg.pct_of_peak_sustained_elapsed (>60%)
- launch__grid_size

Rules:
- 1D: (cdiv(N, BLOCK))
- 2D: (cdiv(M, BLOCK_M), cdiv(N, BLOCK_N))
- 3D: (batch, cdiv(M, BLOCK_M), cdiv(N, BLOCK_N))
- >3D: flatten ONLY independent dims
- Prefer batch / head / expert / group parallelism before shrinking BLOCK
- For grouped operations: ensure group dimension is in grid (e.g., program_id(2) for
    groups)
- Change grid only if SM utilization is clearly low

Safety:
- Max 3 grid dims, static rank
- grid=(G0,G1,G2) must match tl.program_id(0/1/2)
- For grouped ops: verify group indexing is correct
- If unsure about correctness, do NOT change grid

Autotune:
- Autotune either BLOCK_* OR (num_warps, num_stages)
- If autotuning BLOCK_*, use grid=lambda META: (...)
- Never redefine BLOCK_* in both kernel and launch
- Max 2-3 configs to reduce compilation time
```

### E.2.2. TENSOR TILING

This stage optimizes the granularity of computation within each thread block by selecting appropriate BLOCK_M, BLOCK_N, and BLOCK_K sizes. The prompt enforces power-of-2 constraints and guides autotuning across a small set of configurations to balance data reuse with register pressure.

```
Focus: BLOCK_M/N/K selection.

Metrics:
- sm__warps_active.avg.pct_of_peak_sustained_active (>50%)
```

```
Rules:
- BLOCK_* must be powers of 2
- Tensor Core: BLOCK_M/N multiple of 16, BLOCK_K multiple of 8 (preference)
- FP32: M/N in {32,64,128,256}, K in {16,32,64}
- Avoid oversized tiles (mask waste)
- Keep baseline tile if unsure

Autotune:
- Max 2-3 configs to reduce compilation time
- Autotune ONLY on @triton.jit kernel
""",
```

### E.2.3. MEMORY AND TUNING

The final stage refines memory access patterns and pipeline execution. The prompt directs the agent to tune `num_warps` and `num_stages` based on occupancy and memory stall metrics, while keeping grid configuration and block sizes fixed from previous stages.

```
Focus: Memory optimization and final parameter tuning.

Metrics:
- dram__throughput.avg.pct_of_peak_sustained_elapsed
- lts__t_sector_hit_rate.pct
- smsp__warp_issue_stalled_memory_dependency_per_warp_active.pct (<20%)
- sm__warps_active.avg.pct_of_peak_sustained_active

Parameters to tune:
- num_stages in {2, 3, 4}
- num_warps in {4, 8} (based on occupancy)

Rules:
- Increase num_stages only if memory stalls > 20%
- Change num_warps only if occupancy suggests it
- Larger BLOCK_K improves reuse but increases register pressure
- Do NOT modify grid or BLOCK sizes (fixed in earlier stages)
- Do not rewrite access patterns without metric evidence

Autotune:
- Max 3-4 configs combining num_stages and num_warps
- Always include original config as baseline
- Revert if gain < 2% or unstable
"""
```

### E.3. Profile Agent Prompts

The profile agent analyzes kernel performance and identifies optimization opportunities. At different stages, it receives different system prompts to align with stage-specific objectives.

### E.3.1. ALGORITHMIC REFINEMENT STAGE

During algorithmic refinement, the profile agent receives the PyTorch reference code, current Triton kernel implementation, NCU profiling metric, and stage-specific prompts as input. The prompt guides the agent to analyze high-level algorithmic optimizations (operator fusion, algorithm replacement, etc.) through structured analysis: code inspection, performance diagnosis, and root cause identification. It outputs a JSON response specifying the identified bottleneck and the modification plan for structural transformation.

```
You are a GPU kernel optimization architect. Analyze the kernel and identify **ONE
    high-level algorithmic optimization**.
```

```
# PyTorch Reference
```python
$python_code
```

# Current Triton Kernel
```python
$triton_code
```

# Nsight Compute Metrics
```
$NCU_METRICS
```

## Analysis Steps

1. **Code Analysis**: Count kernels, identify operations, check for inefficiencies
2. **Performance Diagnosis**: Use metrics/latency to identify bottleneck type
3. **Root Cause**: Combine code + performance to find the core issue

## Optimization Categories (pick ONE if worth optimizing):

### 1. Operator Fusion
Fuse consecutive ops into fewer kernels to reduce memory traffic and launch overhead.

### 2. Algorithm Replacement
Replace naive algorithm with optimized variant.
- For Attention: Flash Attention, online softmax
- For Convolution: Winograd, im2col
- For RNN/GRU/LSTM: Persistent kernel with HYBRID computation

### 3. Kernel Launch Reduction
Combine multiple small kernels to reduce overhead.

### 4. Memory Layout Optimization
Use in-place operations, buffer reuse, or better layouts.

## Should We Optimize?

Before proposing optimization, determine if it's worthwhile:
- **Not worth optimizing** if:
  - Code is already near-optimal (expected speedup < 10%)
  - Bottleneck cannot be addressed (hardware limited, already optimal algorithm)
  - Optimization would add significant complexity with minimal gain

- **Worth optimizing** if:
  - Clear algorithmic inefficiency exists (multiple kernels, suboptimal algorithm)
  - Expected speedup >= 20%
  - Concrete optimization path available

## Output (JSON)

```json
{
  "worth_optimizing": "yes/no",
  "bottleneck": "<Root cause in 1-2 sentences>",
  "modification plan": "<Implementation steps in 2-3 sentences>",
}
```
Return JSON only.
```

E.3.2. HARDWARE-SPECIFIC TUNING STAGES

During hardware-specific tuning, the profile agent receives the PyTorch reference code, current kernel code, stage-specific system prompts, and NCU profiling metrics as input. The prompt constrains analysis to the current stage's optimization scope (e.g., grid configuration, block sizes, or memory patterns) and directs the agent to propose exactly one targeted modification based on measured bottlenecks. It outputs a JSON response specifying the bottleneck and modification plan.

```
You are a senior Triton kernel optimization engineer. Read the PyTorch reference
    code, the current Triton candidate, and the Nsight Compute metrics. Then identify
    one highest-impact speed bottleneck, propose one optimization method and propose
    a modification plan. Be surgical and metrics-driven.

# PyTorch Reference
```python
$python_code
```

# Current Triton Kernel
```python
$TRITON_CODE
```

# Current Optimization Stage
```
$STAGE_CONTEXT
```

# Nsight Compute Metrics
```
$NCU_METRICS
```

Rules:
- Return **one** optimization method -- the largest expected speedup.
- Focus on Triton-specific optimizations:
   * **BLOCK_M/N/K tuning**: Adjust tile sizes to optimize data reuse and cache
     efficiency
   * **num_warps**: Control occupancy (2/4/8 warps per block)
   * **num_stages**: Enable software pipelining (2-4 stages for memory-bound kernels)
   * **Memory access patterns**: Optimize coalescing, use tl.trans() for layout changes
   * **Grid configuration**: Adjust program_id mapping and workload distribution
- Prefer changes that directly address measured bottlenecks from NCU metrics:
   * High DRAM throughput -- Increase BLOCK size for data reuse
   * Low cache hit rate -- Adjust BLOCK size for better locality
   * Low occupancy -- Tune num_warps, reduce register pressure
- Keep fields brief; avoid lists of alternatives, disclaimers, or generic advice.

Output format (JSON):
```json
{
  "bottleneck": "<max 30 words>",
  "modification plan": "<max 35 words>"
}```
Return JSON only.
"""
```

## E.4. Code Agent Prompts

The code agent generates or modifies kernel implementations based on feedback from the profile agent or the debug agent. Different prompts guide optimization or repair tasks.

E.4.1. OPTIMIZATION MODE (ALGORITHMIC REFINEMENT)

During algorithmic refinement, the code agent receives the current kernel and optimization analysis as input. The prompt directs the agent to implement structural transformations suggested by the profile agent and apply specific algorithmic changes while preserving correctness. It outputs the optimized Triton kernel code.

```
You are optimizing a Triton kernel based on algorithmic analysis.

# Current Kernel (needs optimization)
```python
$current_kernel
```

# Analysis Results
```
$ANALYSIS_INFO
```

# Your Task

Implement the optimization strategy above. Focus on the specific bottleneck
    identified.

## Key Requirements

1. **Preserve correctness**: Maintain the same input/output behavior
2. **Apply the optimization**: Follow the implementation plan exactly
3. **Use valid Triton syntax**:
   - Every kernel MUST have `@triton.jit` decorator
   - Grid size MUST be > 0: use `triton.cdiv(N, BLOCK)` or `max(1, N // BLOCK)`
   - BLOCK sizes MUST be power-of-2: 16, 32, 64, 128, 256
   - No `continue`, `break`, `return` inside kernels (use masking)
   - Prefer `tl.dot(a, b, allow_tf32=True)` for matmul operations
4. **Output format**:
   - Imports: `import torch, torch.nn as nn, triton, triton.language as tl`
   - `@triton.jit` kernel(s)
   - Wrapper function(s)
   - `class ModelNew(nn.Module)` -- REQUIRED
   - NO testing code, NO `if __name__ == "__main__"`

Do NOT include: testing code, if __name__, get_inputs, get_init_inputs

```python
# <optimized Triton code>
```
"""
```

E.4.2. OPTIMIZATION MODE (HARDWARE-SPECIFIC TUNING)

During hardware-specific tuning stages, the code agent receives the current kernel, stage-specific system prompts, and optimization analysis as input. The prompt directs the agent to apply focused hardware-level optimizations based on the profile agent's suggestions. It outputs the optimized Triton kernel code.

```
You are a Triton kernel optimization specialist. Generate the FASTEST possible kernel
    based on the analysis.

# Target GPU:
$gpu_name

# Current Kernel (needs optimization)
```python
```

```
$current_kernel
```

# Current Optimization Stage
```
$STAGE_CONTEXT
```

# Analysis Results
```
$ANALYSIS_INFO
```

## CRITICAL -- Code MUST compile and run:
1. EVERY kernel function MUST have `@triton.jit` decorator
2. Grid size MUST be > 0: use `triton.cdiv(N, BLOCK)` or `max(1, N // BLOCK)`
3. BLOCK sizes MUST be power-of-2: 16, 32, 64, 128, 256
4. `tl.program_id(axis)` only supports axis = 0, 1, 2
5. No `continue`, `break`, `return` inside loops -- use masking
6. No tensor indexing with loop vars: `x[:, i]` is INVALID
7. mask shape MUST match data shape in tl.load/tl.store

## Missing Triton Functions (implement manually):
- tl.tanh, tl.sigmoid, tl.gelu, tl.silu, tl.softmax, tl.mish

## OUTPUT FORMAT (STRICT):
1. Imports: torch, torch.nn, triton, triton.language as tl
2. @triton.jit decorated kernel function(s)
3. Wrapper function(s) for grid calculation and kernel launch
4. class ModelNew(nn.Module) that calls your kernels

Do NOT include: testing code, if __name__, get_inputs, get_init_inputs

```python
# <optimized Triton code>
```
```

### E.4.3. REPAIR MODE

When kernel execution fails, the code agent receives kernel code and debug analysis as input. The prompt emphasizes strict adherence to Triton syntax rules and output format requirements. It outputs corrected Triton kernel code.

```
Fix the Triton kernel errors. Generate correct code based on the error analysis.

# Broken Code
```python
$OLD_CODE
```

# Analysis Results
```
$ANALYSIS_INFO
```

## OUTPUT FORMAT (STRICT):
1. Imports: torch, torch.nn, triton, triton.language as tl (and math if needed)
2. @triton.jit decorated kernel function(s)
3. Wrapper function(s) for grid calculation and kernel launch
4. class ModelNew(nn.Module) -- REQUIRED

Do NOT include: testing code, if __name__, get_inputs, get_init_inputs

```

```python
# <corrected code>
```

## E.5. Debug Agent Prompt

The debug agent receives error logs, PyTorch reference implementations, and broken kernel code as input. The prompt guides the agent to diagnose execution failures and identify root causes with specific, actionable diagnostic information. It outputs a structured JSON response containing the critical issue and required modifications, which the code agent then uses to generate fixes.

```
You are a Triton kernel debugging expert. Analyze the error and identify the root
    cause.

# Error Log
```
$ERROR_LOG
```

# Expected Behavior (PyTorch Reference)
```python
$PYTORCH_CODE
```

# Current Implementation (Broken Triton Kernel)
```python
$KERNEL_CODE
```

# Your Task

Identify the **most critical issue** that causes the error above.

## Analysis Guidelines

1. **Focus on root cause**, not symptoms
   - Bad: "Output is wrong"
   - Good: "BLOCK_K loop missing, only processes first 32 elements of K dimension"

2. **Be specific about WHAT and WHERE**
   - Bad: "Memory access issue"
   - Good: "Line 45: tl.atomic_add(c_block_ptr, acc) - atomic_add requires scalar
    pointer, not block_ptr"

3. **Prioritize by impact**
   - Correctness bugs > Performance issues > Style problems
   - Algorithm errors > Implementation details

## Output Format
```json
{
  "critical_issue": "<Concise description of THE root cause, max 30 words>",
  "modification plan": "<What needs to change (not how), max 30 words>"
}
```
Return JSON only.
```

# F. Nsight Compute Profiling Metrics

NVIDIA Nsight Compute (NCU) provides detailed hardware-level performance metrics for analyzing GPU kernel execution. In our framework, NCU profiling is invoked at each optimization stage to provide low-level performance signals that guide

the profile agent's analysis and modification planning.

We collect a unified set of core metrics covering compute utilization, memory hierarchy behavior, and execution stalls. The following list shows the metrics reported to the profile agent during each profiling step:

```
"sm__throughput.avg.pct_of_peak_sustained_elapsed",   # SM compute utilization
"launch__grid_size",                                  # Global grid size
"sm__warps_active.avg.pct_of_peak_sustained_active",  # Warp occupancy
"dram__throughput.avg.pct_of_peak_sustained_elapsed", # DRAM bandwidth utilization
"lts__t_sector_hit_rate.pct",                         # L2 cache hit rate
"smsp__warp_issue_stalled_memory_dependency_per_warp_active.pct",  # Memory
    dependency stalls
```

Although the same metric set is collected throughout all stages, different optimization stages emphasize different performance aspects according to their objectives:

- **Algorithmic refinement:** SM utilization, global parallelism (grid size), and DRAM bandwidth utilization.

- **Parallel mapping:** warp occupancy, global parallelism (grid size), and SM utilization.

- **Tensor tiling:** DRAM throughput, L2 cache hit rate, and memory-dependency stalls.

- **Memory optimization:** memory-dependency stalls, DRAM throughput, and L2 cache hit rate.

By providing stage-specific metrics to the profile agent, we enable focused bottleneck diagnoses and targeted modification plans.

# G. Limitations

**Dependence on Expert Optimization Priors.** EGG relies on a set of expert-designed optimization principles to guide the staged search process. While these priors substantially improve search efficiency, they may also bias exploration toward known optimization patterns and limit the discovery of unconventional but potentially superior designs. Enabling the framework to automatically learn or adapt expert priors remains an important direction for future work.

**Lack of Joint Cross-Stage Optimization.** EGG decomposes optimization into sequential stages to restrict the search space, improving controllability and stability in the open-ended LLM generation setting. However, this design limits cross-stage interactions: locally optimal decisions at individual stages may not always lead to the best final performance when combined. For example, the best parallelization choice under one tiling strategy may be worse than another parallelization choice paired with a different tiling configuration. We make this tradeoff because jointly modifying multiple coupled factors enlarges the effective proposal space and increases search cost. Compared with the search-based baseline CudaForge, EGG achieves better performance with a smaller budget (50k vs. 110k output tokens per kernel). A promising future direction is to propagate top-$k$ candidates after each stage when a larger token budget is available, enabling limited joint exploration while reducing the risk of discarding candidates that may improve in later stages.

