# OpenReview forum: "EGG: An Expert-Guided Agent Framework for Kernel Generation"
_ICML.cc/2026/Conference — ICML 2026 regular_

### Official Review · Reviewer_LSW6 · 2026-03-12

**Soundness:** 3
**Presentation:** 3
**Significance:** 2
**Originality:** 2
**Overall Recommendation:** 4
**Confidence:** 4

**Summary:**

This paper proposes an expert-designed workflow for automated GPU kernel generation consisting of two stages: algorithmic selection/refinement and hardware-specific tuning. A multi-agent system coordinates profiling, code generation, and debugging during the kernel development process. The method is evaluated on KernelBench and three representative kernels from TritonBench. The reported results show an average 2.13× speedup over PyTorch baselines, suggesting that the proposed workflow can effectively improve kernel performance.

**Compliance With Llm Reviewing Policy:**

Affirmed.

**Key Questions For Authors:**

(1) In the algorithm refinement stage, only the variant with the highest measured speedup is passed to the hardware-specific tuning stage. Could this early filtering miss opportunities where a currently slower algorithm becomes superior after hardware-level tuning?

(2) What is the repair success rate of the error-guided kernel fixing process for different error types (e.g., syntax errors vs. runtime errors)? A detailed breakdown would help better understand the limits of LLM-based kernel repair.

**Limitations:**

Yes, in appendix

**Strengths And Weaknesses:**

### Strengths

(1) Expert-designed stage decomposition allows the LLM to focus on smaller and simpler sub-tasks, significantly reducing the complexity compared to one-shot kernel generation.

(2) Error-message-guided and profiling-guided kernel rewriting effectively improves both the correctness rate and the final performance of generated kernels.

### Weaknesses

(1) The pipeline keeps only the single best-performing kernel before entering the hardware-specific tuning stage. It is unclear whether this early filtering may discard kernels with high optimization potential that could outperform the selected candidate after further tuning.

(2) Although the pipeline appears conceptually language-independent, the paper only demonstrates it using Triton. How the approach can be extended to other kernel languages such as CUDA, and what modifications would be required, is not discussed.

---

> ### Author Rebuttal · Authors · 2026-03-31
>
> **1. Early filtering may miss late-blooming candidates**
>
> This is a very reasonable concern. Some candidates that look weak early may become strong after later hardware-specific tuning. It is a practical tradeoff among token cost, search breadth, and performance. In the current LLM-based setting, correctness and latency already provide a strong first-order signal, so we use them to prune clearly low-value candidates before entering the much more expensive later tuning stages. Compared with the search-based baseline CudaForge, EGG achieves better performance with smaller budget (50k vs. 110k output tokens per kernel), suggesting that the current design is effective in practice. Our framework can also be naturally extended to keep top-k candidates, which would reduce the risk of discarding late-blooming solutions to further improve performance, at the cost of a larger token budget.
>
> **2. Transferability beyond Triton**
>
> At the framework level, EGG is not Triton-specific; what is Triton-specific is the current instantiation layer, including prompts, tuning knobs, and compiler/runtime hooks. Please see our response to Reviewer 3jc5 (R2), Q3 for details.
>
> **3. Repair success rate by error type**
>
> We analyzed the original evaluation logs from Level 1 to Level 3 by tracking each candidate's first failure type and whether it was later repaired into a runnable kernel. Overall, the debug agent is most effective for implementation and compilation errors (e.g., missing symbols and syntax errors), with repair rates close to 95% in our current sample. The more challenging cases are numerical mismatches, especially in complex fused kernels from Level 2 and Level 3 (repair rates 75%). Representative examples include Level 2 id=62 (`Matmul -> GroupNorm -> LeakyReLU -> Sum`) and id=95 (`Matmul -> Add -> Swish -> Tanh -> GELU -> Hardtanh`). In such cases, LLMs often struggle to diagnose and fix numerical mismatches caused by fusing multiple numerically sensitive computations. When repeated repair attempts still fail, the model may naturally fall back to smaller-granularity fusion. Overall, LLM-based kernel repair is highly effective for local implementation and compilation errors, while its main current limitation lies in resolving numerical mismatches in complex fused kernels.

---

> > ### Author Rebuttal · Reviewer_LSW6 · 2026-04-03
> >
> > For 2, I am still skeptical about the framework’s generalizability. CUDA exposes much lower-level details than Triton, so whether a direct instantiation would work remains unclear. What’s missing is evidence that the decomposition and LLM-driven decisions remain valid under a much larger and less structured optimization space, rather than requiring fundamental changes to the framework.

---

> > > ### Author Response · Authors · 2026-04-04
> > >
> > > Thank you for the follow-up. To address this concern directly, we implemented a CUDA instantiation of EGG by replacing the prompts and compiler/runtime interfaces with CUDA-based counterparts. On 15 representative Level-2 fused operators spanning Conv2d, ConvTranspose, and GEMM fused pipelines, EGG achieves 100% correctness, 100% Fast1, and 2.38x average speedup over PyTorch. These results provide direct evidence that EGG’s framework-level decomposition and stage-wise decision process transfer beyond Triton without requiring a fundamentally different design.
> > >
> > > The key issue, in our view, is not whether Triton-specific components can be simply replaced, but whether EGG’s decomposition and LLM-driven decisions remain effective in CUDA’s larger and less structured optimization space. Since CUDA does not provide the same built-in structure as Triton (e.g., BLOCK_M/N/K, num_warps, num_stages, tl.dot), we explicitly impose structure in two ways: 1) during seed generation, we provide few-shot CUDA exemplars for core computations, such as GEMM and reduction/pooling, instead of asking the model to generate arbitrary CUDA code from scratch; 2) during algorithmic refinement, the model must first determine the fusion/rewrite boundary and then choose an appropriate CUDA implementation path for the resulting kernel (e.g., CUDA Core vs. Tensor Core). In this way, we impose structural constraints on the candidate space, making it more controllable for the LLM. We will add these results to the final version and clarify that CUDA’s less structured optimization space mainly manifests as greater backend implementation difficulty, rather than a need for a fundamentally different decomposition framework.

---

### Official Review · Reviewer_9Zen · 2026-03-13

**Soundness:** 3
**Presentation:** 3
**Significance:** 3
**Originality:** 3
**Overall Recommendation:** 4
**Confidence:** 4

**Summary:**

This paper presents EGG, an expert-guided agent framework for GPU kernel generation. EGG incorporates expert workflow, consisting of algorithmic structure design and hardware-specific tuning, into the AI-based Triton kernel generation. In the hardware-specific stage, there are three sequential sub-stages (parallel mapping, tensor tiling, and memory optimization) to optimize the implementation. Each stage works with stage-aware multi-agent collaboration, where profile/code/debug agents work together. The evaluation on KernelBench achieves an average speedup of 2.13x, outperforming existing works.

**Compliance With Llm Reviewing Policy:**

Affirmed.

**Final Justification:**

The rebuttal addressed my main concerns about evaluation fairness regarding CudaForge / torch.compile() baseline numbers and token budget, and confirmed the advantages of the proposed method. As other reviewers suggest, the rationale for sequential optimization rather than joint optimization, and its transferability to other hardware/languages, is not perfectly clear, but that does not outweigh the paper's contribution. Hence, I would increase the score to weak accept.

**Key Questions For Authors:**

1. For the baseline results in Table 1, did you evaluate yourselves or use the results reported in other papers? The results of CudaForge appear identical to those reported in its original paper (Table 2 of https://arxiv.org/abs/2511.01884v2), but the original paper's data were evaluated with a different model (OpenAI-o3) and hardware (NVIDIA Quadro RTX 6000), so the evaluation may not be fair.
2. Which mode of the torch.compile did you use in the evaluation (e.g., `default`, `max-autotune`)? Did you pick the one that achieved the best performance?
3. In the evaluation, what is the cost of running each system, in terms of number of tokens, API fees, or latency? Does EGG still have advantages over other methods, even with the same budget?
4. Do you have any particular reason that you chose three stages (parallel mapping, tensor tiling, memory optimization) and in this particular order? Does this cover all the optimization patterns and work better than other orders? In tensor tiling, if it is only about determining the tile size based on the grid dimension, why does it require LLM and not just applying a division?
5. For the algorithmic structure design, how many kernel seeds do you generate? Do you do anything special to increase the diversity of the seeds, and what does the "lightweight performance filter" do? Also, do you think this approach can discover new algorithms like the FlashAttention paper did?

Among these, 1-3 are about the fairness of the evaluation and are most critical to my assessment, and I am happy to increase the score if I can confirm the evaluation is fair. 4 and 5 are clarifying questions to understand the proposal better.

**Limitations:**

Yes

**Strengths And Weaknesses:**

Strengths

1. Incorporating expert optimization principles in GPU kernel writing agets is a reasonable approach and achieves a good speedup on KernelBench.
2. The paper provides an ablation on hardware architecture and a case study.

Weaknesses

1. The evaluation may not be very fair. See my questions below.
2. The presentation could be improved. There are various method names across different abstraction layers, which were hard to follow from the abstract and introduction.
3. Many parts of the algorithms are hard-coded and unclear if it is the universally best composition.

---

> ### Author Rebuttal · Authors · 2026-03-31
>
> **1. Misalign the platform of other SOTA results**
>
> We thank the reviewer for pointing out the fairness issue and apologize for not handling it more carefully in the original submission. In the paper, the originally reported CudaForge results were 96% / 54% / 1.45x on Level 1, 100% / 89% / 2.10x on Level 2, and 96% / 68% / 1.28x on Level 3 (Correctness / Fast1 / Speedup). Following the reviewer's suggestion, we reran the relevant baselines under a matched setup using the same model (GPT-5.1) and the same hardware (RTX 4090). Under this matched setting, CudaForge reaches 100% correctness, but its performance remains limited: Level-1/2/3 Fast1 are 56/90/72, with corresponding speedups of 1.43x/2.00x/1.30x. This suggests that EGG's advantage comes from its expert-guided workflow design. We will revise the final version accordingly to ensure a fair comparison.
>
> **2. torch.compile mode**
>
> In the original submission, we evaluated `torch.compile` under its default mode. Following the reviewer's suggestion, we additionally tested the stronger `max-autotune` mode. Even under this setting, our method still achieves clear performance gains: `torch.compile` (`max-autotune`) reaches 66/1.04, 87/1.51, and 84/1.48 on Level 1/2/3 (Fast1/Speedup), while our method reaches 72/1.83, 100/2.73, and 94/1.52.
>
> **3. Search cost and budget comparison**
>
> Please see our response to Reviewer 3jc5 (R2), Q2. Under the same GPT-5.1 + RTX 4090 setup, EGG uses about 50k tokens and 20 minutes per kernel on average, compared with CudaForge's 110k tokens and 30 minutes. Since both methods use the same model, token usage directly reflects relative API cost.
>
> **4. Why this stage order, and why use an LLM for tiling?**
>
> We choose the order because these decisions have a natural dependency hierarchy: parallel mapping determines how work is distributed across program instances/SMs; tensor tiling then decides the tile handled by each instance; memory optimization finally improves locality, coalescing, and pipelining for an execution structure that is already largely fixed. Reversing the order can invalidate earlier decisions: e.g., changing the parallel mapping after tiling may change how output tiles are distributed across SMs, which can reduce occupancy and make the previously chosen suboptimal. We do not claim this order is theoretically optimal for all kernels, but it provides a more stable and practical optimization trajectory for the current LLM-based setting. As shown in Figure 4, each stage delivers additional gains over the previous one, indicating that this ordering yields a stable and cumulative optimization trajectory in practice.
>
> For tensor tiling, the grid does not uniquely determine a good tile shape; one must also consider tradeoffs in register pressure, occupancy, memory access, and tensor-core utilization. This is why a simple division rule is often insufficient. In EGG, the role of the LLM is to propose tiling choices by jointly considering code structure, hardware constraints, and profiling feedback, which often leads to more promising candidates. Figure 4 further shows that this stage contributes an additional 1.08x average performance gain.
>
> **5. Number of seeds, diversity, and algorithm discovery**
>
> In our evaluation, we generate 2 seeds per task from the same seed prompt template, with diversity coming mainly from independent sampling and stochastic decoding; we do not introduce an additional diversity module. The lightweight performance filter is a correctness-plus-latency gate: it first removes non-runnable candidates, then selects the current best candidate from runnable seeds and algorithmically refined candidates for later stages. Since correctness and latency already provide a strong first-order signal, we use them to prune clearly low-value candidates before entering the more expensive later stages. This is a practical tradeoff among search cost, search breadth, and final performance.
>
> Regarding structural discovery, we believe Level-2 id=42 is a representative example. Starting from a `ConvTranspose2d -> GlobalAvgPool` reference, EGG rewrites the computation by exploiting the algebraic structure between the two operators: rather than materializing the full transposed-convolution output, it first aggregates spatially and then contracts with convolution weights summed over spatial dimensions. This avoids explicit generation and memory traffic of the intermediate feature map and achieves 18.92x speedup. This suggests that EGG can go beyond tiling or launch-parameter tuning and actively change computation paths to realize nontrivial structural rewrites. This also gives us new inspiration for future work, where we plan to further explore whether this capability can lead to broader forms of algorithmic discovery and additional performance gains.

---

> > ### Author Rebuttal · Reviewer_9Zen · 2026-04-03
> >
> > Thanks for the rebuttal. That resolves my concerns about evaluation fairness.

---

> > > ### Author Response · Authors · 2026-04-05
> > >
> > > We truly appreciate your kind reassessment and are glad that our rebuttal has resolved your concerns. Thank you also for your thoughtful review and the time you devoted to evaluating our manuscript.

---

### Official Review · Reviewer_3jc5 · 2026-03-13

**Soundness:** 3
**Presentation:** 3
**Significance:** 3
**Originality:** 3
**Overall Recommendation:** 4
**Confidence:** 3

**Summary:**

This paper proposes EGG, an expert-guided multi-agent framework for automatic GPU kernel generation. The method decomposes kernel optimization into two hierarchical stages: algorithmic structure design and hardware-specific tuning. It further introduces stage-aware collaboration among code, profiling, and debugging agents so that optimization decisions are carried across stages in a structured way. On KernelBench and several real-world Triton workloads, EGG reports 100% correctness, a 2.13x average speedup over PyTorch Eager, and stronger results than general-purpose LLMs, AutoTriton, CudaForge, and Torch Compile.

**Compliance With Llm Reviewing Policy:**

Affirmed.

**Key Questions For Authors:**

1. Please provide a more detailed reproducibility appendix for prompts, agent interaction budgets, stopping criteria, and failure handling. This would significantly improve confidence that others can validate the approach.
2. How does total search cost compare against CudaForge and AutoTriton when normalized by wall-clock time, model calls, and GPU profiling runs per solved task?
3. Which parts of EGG are Triton-specific, and which parts are likely to transfer to CUDA C++, ROCm, or compiler-based backends?
4. Did you observe any classes of kernels where the expert-guided staged decomposition systematically hurts exploration compared with a less constrained search policy?

**Limitations:**

yes

**Strengths And Weaknesses:**

This is a good paper overall. The problem is important, timely, and technically challenging. The central design choice, namely constraining search with expert kernel-optimization structure rather than relying on unconstrained trial and error, is well motivated and matches how human experts actually work. The staged decomposition is clear, and the multi-agent design is not just decorative: the ablation study suggests that multi-seed search, algorithmic refinement, hardware tuning, and agent collaboration all contribute materially. The empirical section is also compelling. KernelBench is an appropriate benchmark, the comparison set is strong, and the case study and real-world TritonBench validation make the results easier to trust. I also appreciate that the paper reports correctness-oriented metrics, not only speed.
My main reservations are about reproducibility and scope. The system appears to depend heavily on proprietary LLMs and a fairly elaborate agent orchestration pipeline, so it is hard to judge how easy the method would be to reproduce or adapt. The paper also reports average generation time and token usage, but a more systematic accounting of search cost would be useful, especially relative to baselines like CudaForge. In addition, while the results are strong on NVIDIA GPUs and Triton kernels, it remains unclear how much of the framework transfers to other hardware backends or non-Triton compilation targets. These are real limitations, but they do not outweigh the overall strength of the empirical and conceptual contribution.

---

> ### Author Rebuttal · Authors · 2026-03-31
>
> **1. Reproducibility details**
>
> We thank the reviewer for this suggestion. We agree that reproducibility would be improved by explicitly reporting interaction budgets, stopping criteria, and failure recovery. Since the rebuttal period does not allow appendix expansion, we will add a dedicated reproducibility appendix to the final version.
>
> The current system uses a fixed-budget pipeline. For each task, we generate 2 seeds, and each seed may be repaired 3 times. If no seed becomes runnable, the task terminates early. For each runnable seed, we run 1 algorithmic analysis. If the analysis returns "not worth optimizing", we keep the original seed. Otherwise, we perform 1 algorithmic refinement and, if needed, 1 repair. We then keep the current best candidate and run three hardware-tuning stages in order: parallel mapping, tensor tiling, and memory optimization. Each stage allows 1 optimization and, if needed, 1 repair. If a stage does not improve performance, we keep the previous best candidate.
>
> Failure handling is also explicit. Validation is always run in isolated spawned subprocesses to avoid CUDA-context contamination. Compilation/runtime failures and accuracy failures are sent to the debug agent for repair. NCU profiling failure is treated as non-fatal and later stages still proceed. If an LLM output is malformed or truncated, the system retries once; if the attempt still fails, it retains the current best candidate.
>
> **2. Search cost and budget comparison**
>
> Compared with the search-based baseline CudaForge, EGG achieves stronger performance with a smaller average budget under the same experimental setup (GPT-5.1 on RTX 4090): CudaForge uses about 110k tokens and 30 minutes per kernel on average (under the same 10-search-iteration setting reported in the paper), whereas EGG uses about 50k tokens and 20 minutes per kernel under our fixed staged optimization budget. This suggests that EGG uses the inference budget substantially more efficiently, and we expect it to remain more favorable under a matched budget. A key reason is that EGG decomposes optimization into expert-guided stages with explicit objectives, which constrains the proposal space and reduces ineffective search.
>
> AutoTriton is less directly comparable in search-cost accounting because it relies on a fine-tuned model with local deployment and is essentially a one-shot generator rather than an agentic search system.
>
> **3. Transferability beyond Triton**
>
> At the method level, EGG is not Triton-specific. Its transferable components include the four-stage optimization decomposition, stage-aware agent coordination and context management, and the feedback loop driven by profiling, correctness, and runtime signals. These are general optimization principles rather than Triton-specific abstractions. What is currently tied to Triton is the instantiation layer: the prompts encode Triton syntax, the tuning stage targets Triton-specific knobs such as `BLOCK_M/N/K`, `num_warps`, `num_stages`, and `program_id` mapping, and the execution/validation pipeline is built around Triton's compilation toolchain.
>
> Extending EGG to other backends would keep the core workflow unchanged while replacing backend-specific interfaces. For CUDA, this mainly requires CUDA-oriented prompts, tuning parameters, and compiler/runtime hooks; the same principle applies to ROCm/HIP. Our intended claim is therefore that EGG is backend-transferable as a framework; we validate Triton first because it is more structured than native CUDA C++, making it a natural starting point for LLM-guided kernel generation. We will clarify this in the appendix and more explicitly distinguish framework-level generality from the current Triton-based instantiation.
>
> **4. Staged decomposition may underperform search**
>
> Yes. We observed this most clearly for kernels combining a large GEMM with a normalization- or reduction-heavy epilogue, where the key challenge is choosing the right fusion boundary. A representative case is Level-2 id=37 (`Matmul -> Swish -> BiasAdd -> GroupNorm`). Here, EGG's algorithmic refinement fused the full pipeline into a single kernel, while the search-based baseline CudaForge, after several rounds, shifted to a different boundary: GEMM remained separate, and Swish + Bias + GroupNorm were fused into a specialized CUDA kernel. That alternative decomposition achieved higher speedup. This suggests that expert-guided staging can sometimes lock in the algorithmic structure too early.
>
> At the same time, decomposing optimization into stages with explicit objectives is intended to make performance improvement more stable. In practice, EGG achieves better average performance than CudaForge under a smaller budget. This suggests that the current design is effective in practice. Besides, our framework can also be naturally extended to keep top-k candidates, which provides a straightforward way to further improve final performance at the cost of a larger token budget.

---

> > ### Author Rebuttal · Reviewer_3jc5 · 2026-04-08
> >
> > Thanks for your rebuttal. My concerns have been resolved.

---

### Official Review · Reviewer_oTvL · 2026-03-15

**Soundness:** 2
**Presentation:** 2
**Significance:** 3
**Originality:** 2
**Overall Recommendation:** 3
**Confidence:** 4

**Summary:**

EGG proposes an LLM-based agent framework for automated GPU kernel generation. The key idea is to decompose kernel optimization into two hierarchical stages, algorithmic structure design and hardware-specific tuning, mirroring expert workflows. A stage-aware multi-agent collaboration mechanism (code, profile, and debug agents) coordinates optimization within and across stages. Experiments on KernelBench and real-world workloads report a 2.13× average speedup over PyTorch Eager.

**Compliance With Llm Reviewing Policy:**

Affirmed.

**Key Questions For Authors:**

Seed Generation Granularity. How are seeds obtained during algorithmic structure design? For a complete network containing many operators, does EGG generate seeds for every individual operator separately? How are operator boundaries defined, and who determines the granularity of fusion? Taking the FlashAttention example, is the fused attention kernel provided as a manually crafted seed, or can EGG automatically derive it from a naive multi-kernel attention implementation?

Rationale for Multi-Agent Design. What is the concrete benefit of using multiple agents over a single agent with separate prompts? If all agents share the same underlying model without task-specific fine-tuning, is the multi-agent design essentially equivalent to maintaining separate conversation contexts? The paper should clarify whether there is a structural benefit beyond context isolation.

Novelty of Structured Context Management. The inter-stage context propagation strategy, retaining finalized decisions and discarding intermediate outputs, appears to be a straightforward engineering choice rather than a novel technical contribution. Could the authors clarify what is specifically novel about this mechanism beyond standard prompt engineering practice?

Computational Cost. How long does kernel generation take per task? The paper reports approximately 20 minutes and 50,000 output tokens per kernel, but it is unclear how this compares to torch.compile() in practical deployment settings. Is EGG intended for offline kernel library generation, or is it expected to run at deployment time?

Underperformance Relative to PyTorch Eager. Table 1 shows that 28% of Level 1 tasks do not outperform PyTorch Eager. What causes this? Additionally, many LLM-based baselines show very low Fast1 rates, suggesting that LLMs can actively degrade kernel performance. What is the underlying cause of this degradation, and how does EGG avoid it?

**Limitations:**

EGG presents a well-organized system with strong empirical results on correctness and speedup. However, the limited originality of the core decomposition idea, the lack of justification for sequential over joint optimization, and the absence of non-LLM baselines are significant concerns that weaken the contribution. Addressing these points would substantially strengthen the paper.

**Strengths And Weaknesses:**

Strengths
Reasonable Stage Decomposition. Rather than applying LLMs naively to end-to-end kernel generation, EGG structures the optimization process into stages, algorithmic refinement, parallel mapping, tensor tiling, and memory optimization, each of which corresponds to a well-understood performance lever in GPU kernel design. This staged decomposition provides clearer optimization objectives at each step and is a reasonable design choice compared to unstructured trial-and-error approaches.

Practically Valuable Direction. Automated kernel generation for deep learning workloads is an important and timely problem. Leveraging coding-specialized LLM agents for this purpose is a promising direction with significant room for further development, and the empirical results demonstrating consistent correctness (100% success rate) are encouraging.

Clear Writing and Presentation. The paper describes EGG's workflow clearly, stage by stage, and the appendix including prompt details and case studies is helpful for understanding the system's implementation.

'Weaknesses
Limited Originality in Core Idea. The central contribution, decomposing kernel optimization into hierarchical stages, is not new. The specific stages adopted by EGG directly correspond to well-established decompositions in prior non-LLM work. Algorithmic refinement has been addressed by TASO (SOSP'19), PET (OSDI'21), and FlashTensor (PPoPP'25); hardware-specific tuning has been extensively studied in TVM (OSDI'18) and Ansor (OSDI'20). The paper does not sufficiently articulate what originality EGG offers beyond applying LLMs within a decomposition framework that already exists in the literature.

Sequential Stage Independence May Be a Fundamental Limitation. EGG independently optimizes each stage in sequence, but a growing body of recent work argues that this approach is suboptimal and proposes joint multi-level optimization instead. Mirage (OSDI'25) jointly optimizes algorithmic refinement and parallel mapping, while Welder (OSDI'23) and ASPEN (NeurIPS'23) jointly optimize tensor tiling and memory access. The paper does not adequately engage with this line of work or justify why sequential independent optimization is the right design choice given these alternatives. The authors themselves acknowledge this as a limitation in the appendix, but the implications for the validity of the overall approach deserve more discussion in the main text.

Insufficient Experimental Baselines. The evaluation compares EGG exclusively against other LLM-based kernel generation approaches. Comparisons against state-of-the-art non-LLM compilers are missing, which makes it difficult to assess how much the LLM-based approach actually contributes beyond what conventional compilation and search methods already achieve. Specifically, comparisons against Mirage (OSDI'25) and Relax (ASPLOS'25) would significantly strengthen the empirical case. Additional hardware coverage beyond RTX 4090 and RTX 5090, such as H100, B200, or RTX Pro 6000, would also improve the generalizability of the results.

---

> ### Author Rebuttal · Authors · 2026-03-31
>
> **1. Originality of the core idea**
>
> Our contribution is not staged optimization itself, but how to make expert kernel-optimization knowledge executable for LLM-based kernel generation so that correctness and performance can be improved reliably. This distinguishes EGG from compilers such as TASO and Relax, which optimize within predefined transformation rules or template-based search spaces. In contrast, EGG targets an open-ended LLM generation setting, where the search space is implicit and optimization lacks stable trajectories. EGG introduces an expert-guided workflow with role-specialized agents and structured feedback, enabling LLMs to progressively apply expert knowledge for kernel synthesis. To our knowledge, EGG is the first framework to combine expert knowledge and LLM agents for kernel generation while achieving SOTA results.
>
> **2. Sequential vs. joint optimization**
>
> We agree that joint optimization is powerful in traditional compilers such as Mirage/Welder, where the search space is explicit and well structured. In our LLM-based generation setting, however, jointly modifying multiple coupled factors enlarges the effective proposal space and search cost. By contrast, staged optimization decomposes this process into subproblems with clearer objectives, yielding more stable gains (Figure 4). Compared with the less structured CudaForge, EGG achieves better performance with fewer output tokens (50k vs. 110k per kernel). It can also be extended to keep top-k candidates after algorithmic refinement and jointly optimize them in later stages.
>
> **3. Baseline completeness and hardware coverage**
>
> We added comparisons with TVM Relax (v0.24) on 25 cases from the three benchmark levels. Overall, EGG achieves 1.56x speedup over TVM and 1.92x over PyTorch. The gap is smaller on Level 3, where TVM benefits from graph-level optimization. EGG shows clearer advantages on Levels 1/2, where its gains come from expert-guided computation restructurings rather than rule-based fusion and scheduling. We also expanded hardware coverage on Level 2. On RTX PRO 6000, EGG achieves 93/2.95 in terms of Fast1/Speedup; on H20, it achieves 88/3.23. We will incorporate these results into the final version.
>
> **4. Seed generation granularity**
>
> Seed kernels are produced from the seed prompt template in Appendix C.1, and diversity comes from independent sampling and stochastic decoding. For networks, EGG does not generate seeds separately, since that would destroy fusion opportunities. Instead, it treats the full target computation as the generation unit, and algorithmic refinement lets the LLM infer suitable fusion granularity. For FlashAttention tasks, we do not provide a fused attention kernel as the seed; the system starts from a naive PyTorch eager reference and lets the LLM propose a fused implementation.
>
> **5. Rationale for multi-agent design**
>
> We agree that context isolation is an important motivation, but its benefit goes beyond that. In long-horizon optimization, giving one agent all history and logs can push it beyond an effective context sweet spot and reduce decision stability. EGG therefore uses multiple agents as a context firewall: each agent works in a smaller context for one sub-objective and communicates through compact structured outputs. The key benefit is role specialization plus controlled information flow, which reduces drift and irrelevant context accumulation. This is supported by Table 2.
>
> **6. Novelty of inter-stage context propagation**
>
> In EGG, this is a key mechanism for stage-aware optimization rather than generic prompt engineering. The main point is not simply to shorten context, but to propagate only stage-validated decisions aligned with the next-stage objective while filtering out exploratory traces, which helps later-stage optimization remain focused, stable, and cumulative.
>
> **7. Computational cost and intended use**
>
> Each kernel takes about 20 minutes with 50k output tokens on average. EGG is intended for offline kernel library generation rather than deployment-time online compilation like `torch.compile()`. Its practical goal is to replace the original implementation at the code level with a more efficient kernel that can be integrated into production inference frameworks such as vLLM. Its cost is an offline engineering cost, mainly from LLM calls, profiling, and verification. Please see Reviewer 3jc5 (R2), Q2 for detailed cost comparisons.
>
> **8. Underperformance relative to PyTorch**
>
> For the Level-1 operators that do not outperform PyTorch Eager, the main reason is that many are already highly optimized, bandwidth-bound kernels such as Sigmoid and GELU, leaving limited headroom for custom kernels. For plain LLM baselines, the low Fast1 reflects both correctness failures and performance regression in one-shot generation. EGG addresses both through multi-agent repair and expert-guided optimization, improving initial seeds into high-performance kernels (Section 4.4).

---

> > ### Author Rebuttal · Reviewer_oTvL · 2026-04-03
> >
> > Thank you for the rebuttal. I'm not sure the idea of LLM-based kernel generation entirely new.
> > https://pytorch.org/blog/kernelfalcon-autonomous-gpu-kernel-generation-via-deep-agents/

---

> > > ### Author Response · Authors · 2026-04-03
> > >
> > > We sincerely appreciate your acknowledgment that our responses to Questions 2-8 (sequential design, baselines, seeds, multi-agent design, cost, etc.) have addressed your concerns. Regarding the remaining issue, which mainly concerns the originality of our method, we provide a more focused clarification below.
> > >
> > > We do not claim that the idea of LLM-based kernel generation is entirely new. Our core contribution is to operationalize explicit expert optimization knowledge as an executable workflow for LLM-based kernel generation, while achieving significant improvements in both correctness and performance.
> > >
> > > The table below summarizes the comparison between EGG and KernelFalcon (method in the link):
> > >
> > > | Method | Expert-guided staged optimization | Algorithmic refinement | Hardware-specific tuning | Execution-based feedback | Profiling-guided feedback | Code/profile/debug role split | Fuser/Extractor/Composer split | Context isolation | Context compression |
> > > |---|---|---|---|---|---|---|---|---|---|
> > > | EGG | ✓ | ✓ | ✓ | ✓ | ✓ | ✓ | ✗ | ✓ | ✓ |
> > > | KernelFalcon | ✗ | Partial | ✗ | ✓ | ✗ | ✗ | ✓ | ✓ | ✗ |
> > >
> > > Overall, the key difference lies in **motivation**: **EGG** is designed to operationalize expert optimization principles so as to constrain the search space and guide LLMs toward improvements in both correctness and performance, whereas **KernelFalcon** primarily uses LLMs to decompose tasks for correctness-oriented synthesis.
> > >
> > > This difference is further reflected in the following aspects:
> > >
> > > - **Optimization structure**
> > >
> > >   **EGG** explicitly decomposes the optimization process into algorithmic refinement and hardware-specific tuning to progressively improve performance. The latter further includes dedicated submodules for parallel mapping, tensor tiling, and memory optimization.
> > >
> > >   **KernelFalcon**’s stages mainly correspond to task decomposition (Fuser, Extractor, KernelAgent, Composer). This is largely covered within EGG's algorithmic refinement stage, while EGG further considers structural rewrites.
> > >
> > > - **Agent design**
> > >
> > >   **EGG** employs stage-aware collaboration (code/profile/debug agent) with structured context propagation (both context isolation and compression).
> > >
> > >   **KernelFalcon** uses stage-specialized agents and parallel workers with isolated execution feedback.
> > >
> > > - **Performance**
> > >
> > >   **EGG** explicitly uses profiling-guided feedback to guide subsequent optimization and reports both 100% correctness and 2.13x performance gains.
> > >
> > >   **KernelFalcon**  primarily validates correctness, with performance analysis left for future work.

---

### Decision · Program_Chairs · 2026-04-30

**Decision:**

Accept (regular)

**Comment:**

The paper proposes an LLM-based agent framework for automated GPU kernel generation. All reviewers agree that the studied problem is important, timely, interesting, and challenging. Also, the proposed method shows good speedup potential. However, reviewers point out that the proposed framework has some problems in lacking discussions on sequential optimization and its transferability to other hardware/languages. Overall, the paper makes some interesting contributions to a challenging problem.